# Initiation of HIV neutralizing B cell lineages with sequential envelope immunizations

Wilton B. Williams[1], Jinsong Zhang [1], Chuancang Jiang[1], Nathan I. Nicely [1], Daniela Fera[2], Kan Luo[1], M. Anthony Moody[1], Hua-Xin Liao[1,8], S. Munir Alam[1], Thomas B. Kepler[3], Akshaya Ramesh[3], Kevin Wiehe[1], James A. Holland[1], Todd Bradley[1], Nathan Vandergrift[1], Kevin O. Saunders[1], Robert Parks[1], Andrew Foulger[1], Shi-Mao Xia[1], Mattia Bonsignori[1], David C. Montefiori[1], Mark Louder [4], Amanda Eaton[1], Sampa Santra[2], Richard Scearce[1], Laura Sutherland[1], Amanda Newman[1], Hilary Bouton-Verville[1], Cindy Bowman[1], Howard Bomze[1], Feng Gao[1], Dawn J. Marshall[1], John F. Whitesides[1,9], Xiaoyan Nie[5], Garnett Kelsoe[1,5], Steven G. Reed[6], Christopher B. Fox[6], Kim Clary[6], Marguerite Koutsoukos[7], David Franco[7], John R. Mascola[4], Stephen C. Harrison[2], Barton F. Haynes[1] & Laurent Verkoczy[1]

A strategy for HIV-1 vaccine development is to define envelope (Env) evolution of broadly neutralizing antibodies (bnAbs) in infection and to recreate those events by vaccination. Here, we report host tolerance mechanisms that limit the development of CD4-binding site (CD4bs), HCDR3-binder bnAbs via sequential HIV-1 Env vaccination. Vaccine-induced macaque CD4bs antibodies neutralize 7% of HIV-1 strains, recognize open Env trimers, and accumulate relatively modest somatic mutations. In naive CD4bs, unmutated common ancestor knock-in mice Env[+]B cell clones develop anergy and partial deletion at the transitional to mature B cell stage, but become Env[−] upon receptor editing. In comparison with repetitive Env immunizations, sequential Env administration rescue anergic Env[+] (non-edited) precursor B cells. Thus, stepwise immunization initiates CD4bs-bnAb responses, but immune tolerance mechanisms restrict their development, suggesting that sequential immunogen-based vaccine regimens will likely need to incorporate strategies to expand bnAb precursor pools.

[1] Duke Human Vaccine Institute, Duke University School of Medicine, Durham, NC 27710, USA. [2] Boston Children's Hospital and Harvard Medical School, Boston, MA 02115, USA. [3] Present address: Boston University School of Medicine, Boston, MA 02118, USA. [4] Vaccine Research Center, National Institute of Allergy and Infectious Diseases, National Institute of Health, Bethesda, MD 20892, USA. [5] Department of Immunology, Duke University, Durham, NC 27710, USA. [6] Infectious Disease Research Institute, Seattle, WA 98102, USA. [7] Present address: GSK Vaccines, Rixensart 1330, Belgium. . [8] Present address: College of Life Science and Technology, Jinan University, Guangzhou 510632, China. [9] Comprehensive Cancer Center, Wake Forest School of Medicine, Winston-Salem, NC 27157, USA. Wilton B. Williams, Jinsong Zhang, Chuancang Jiang, Nathan I. Nicely and Daniela Fera contributed equally to this work. Correspondence and requests for materials should be addressed to B.F.H. (email: Barton.haynes@dm.duke.edu) or to L.V. (email: Laurent.verkoczy@dm.duke.edu)

The HIV-1 envelope (Env) is the target of neutralizing antibodies (nAb)[1]. However, Env-immunogens including stabilized trimers have thus far been ineffective for inducing broadly neutralizing antibodies (bnAbs) in humans or wild-type animals[2–5]. Antibody-virus co-evolution studies from the time of HIV-1 transmission through bnAb development have shown that bnAbs arise after extensive Env diversification; and when bnAbs develop, they are subdominant with respect to other Env lineages[6–8].

BnAb knock-in (KI) mice have proved useful for bnAb development and regulation studies. Several reports with such models have demonstrated that portions of bnAb maturation pathways can be completed by immunization regimens, including: (1) initiation or partial completion of bnAb-like responses with immunogens that target B cell repertoires generated from knocked-in unrearranged bnAb germ line segments[9] or B cells-bearing partially reverted ($V_H$ germ line/mature HCDR3 hybrid) knocked-in rearrangements[10–13] and (2) induction of bnAb responses with immunogens that can engage B cells expressing either near-mature or fully affinity matured bnAb V(D)J rearrangements[12, 14, 15]. However, several mouse models of bnAb development have also demonstrated that bnAb maturation of membrane proximal external region (MPER)-targeting or CD4-mimicking bnAbs[16–18] is likely to be limited at some point in development, either by central or peripheral tolerance controls. We have previously shown that both mature and UCA gp41 MPER bnAb heavy- (HC) and light-chain (LC) gene-rearranged ($V_H DJ_H/V_L J_L$) KI mice have severe bone marrow (BM) deletion, and the few remaining B cells in the periphery are anergic, resulting in massive reduction in BM precursor frequency of MPER bnAbs[16]. Similarly, immunization of rhesus macaques with Env immunogens has initiated bnAb-like lineages that have been controlled either by deletion or affinity reversion (maturation off-target) due to selection of non-bnAb HCDR3 regions[19]. In contrast, the precursor frequency of CD4-mimicking type of CD4-binding site bnAbs (VRC01-class) has been found to be normal in UCA KI mice in one study[9], but abnormal with BM deletion, receptor editing, and peripheral anergy in another[17].

In contrast to the VRC01-class of CD4-binding site bnAbs, the CD4-binding site HCDR3-binder class of bnAbs make contacts with gp120 via their CDR3 loops. CH103, a prototype of the HCDR3-binder class of CD4-binding site bnAbs, is one of the only two bnAb lineages whose complete virus-Ab co-evolution pathway has been comprehensively characterized[6], and whose co-evolved Env maturation pathway, from which sequential immunogens have been derived for this study, can now also be investigated in SHIV CH505-infected non-human primates[20]. No studies have yet been done, however, to characterize the HCDR3-binder-class responses to immunization, nor have any bona fide unmutated common ancestors (UCAs) from full, patient-derived bnAb lineages been studied in the setting of a bnAb KI model. Moreover, the in vivo host controls have yet to be systematically examined in a physiologically relevant setting -- that is, one in which all such controls (including LC receptor editing) are available for the immune system to utilize.

We report here the immunogenicity in rhesus macaques and CH103 CD4-binding site bnAb UCA KI mice of sequential Env immunogens derived from the CH505 HIV-1-infected individual who made the CH103 bnAb lineage. In macaques, vaccine-induced nAbs had epitopes overlapping that of CH103, bound only open trimers, and neutralized rare tier 2 viruses. While the $V_H$ genes encoding vaccine-induced antibodies in macaques were similar to the $V_H$ gene of CH103, the $V_\lambda$ genes were not, raising the possibility of receptor editing. In CH103 bnAb $V_H + V_\lambda$ UCA mice, we found that ~70% of BM UCA B cells were deleted at the transitional to mature B cell development stage, with most of the remaining B cells edited with alternative $V_L$. Sequential gp120 Env immunizations could, however, select for B cells bearing paired CH103 $V_H$ and $V_\lambda$, thereby enlarging the pool of bnAb B cell precursors for further lineage maturation.

## Results

**Immunizations with CH505 4-valent gp120 Env monomers.** We have previously reported isolation from an individual of a CD4-binding site bnAb B cell lineage, CH103, and CH505 Envs that evolved sequentially during the time of bnAb development[6]. From this group of Envs, we generated 113 autologous recombinant Envs and tested them by ELISA for binding to members of the CH103 bnAb lineage[20]. We selected four Envs based on binding to stages of the CH103 bnAb lineage: the CH505 transmitted founder (TF) and three natural CH505 variants (week 53, 78, and 100)[20]. To determine if CH505 sequential vaccine regimens could initiate CH103-like B cell lineages in non-human primates, we vaccinated macaques with CH505 gp120 monomers in the observed temporal order, either sequentially or cumulatively, with the CH505 TF gp120 at all timepoints as control.

CH505 gp120 Envs in AS01E (NHP88, $N = 16$) or GLA-SE (NHP79, $N = 24$) adjuvants and administered either repeatedly as TF Env immunogen alone or as sequential Envs immunogens induced statistically similar plasma-neutralizing antibody profiles in macaques after six immunizations (Fig. 1a–c). However, sequential gp120 Env-immunized macaques showed a trend of qualitatively better serum plasma-neutralizing antibody responses relative to those immunized repeatedly with TF Env gp120, both in terms of potency and breadth (Fig. 1a, b). Furthermore, sequentially-immunized macaques had a trend for enhanced autologous tier 1 neutralization titers relative to TF Env-immunized animals (Fig. 1c), with macaques administered with CH505 TF Env gp120 in GLA-SE ($N = 4$) inducing mean autologous tier 1 (CH505.w4.3)-neutralizing antibody ID50-titers of $614 \pm 597$ (mean $\pm$ SD), vs. those administered with CH505 sequential gp120 Envs in GLA-SE ($N = 4$) inducing mean autologous tier 1 (CH505.w4.3)-neutralizing antibody ID50-titers of $933 \pm 712$. This trend reached near significance ($P = 0.07$, exact Wilcoxon test) in groups administered with CH505 sequential gp120 Envs when they were formulated in AS01E ($N = 8$), with mean autologous tier 1 (CH505.w4.3)-neutralizing antibody ID50-titers of $2264 \pm 2932$, relative to those administered with CH505 TF Env gp120 alone formulated in AS01E ($N = 8$), which by comparison, induced mean autologous tier 1 (CH505.w4.3)-neutralizing antibody ID50-titers of $630 \pm 383$.

The enhanced neutralizing antibody responses in sequentially-immunized groups would be anticipated to be only revealed as a trend in a complete polyclonal system, such as macaques, since any potential enhancements in plasma-neutralizing antibody responses are most likely to be subdominant in nature. Such results therefore predict that general CD4-binding site binding responses will be predominantly non-specific, and thus similar between all immunization groups and adjuvants tested. To examine this, we measured plasma binding in macaques, and as expected, observed no statistical differences or obvious trends in levels of plasma-binding titers to TF Env gp120 (Fig. 1d, e). Specifically, after sixth immunizations, CH505 TF Env gp120 in AS01E ($N = 8$) or GLA-SE ($N = 4$) induced autologous (CH505 TF gp120)-binding-Ab Log AUC-titers of $9.3 \pm 1.0$ and $8.9 \pm 1.6$ ($P = 0.8$, exact Wilcoxon test), respectively, (Fig. 1d) and CH505 sequential Env gp120s in AS01E ($N = 8$) and GLA-SE ($N = 4$) induced autologous Env-binding-Ab Log AUC-titers of $10.2 \pm 0.9$ and $10.0 \pm 0.8$ ($P = 0.7$, exact Wilcoxon test), respectively, (Fig. 1e).

**Vaccine-induced blood memory B cell repertoires in macaques.** To further examine the quality of vaccine-induced antibodies

targeting the CD4-binding site, we compared the binding characteristics of plasma with those of CD4-binding site CH103 early-lineage members. We previously reported that the CH103 UCA bound CH505 TF gp120 Envs, but not Env variants with a deletion of isoleucine at amino acid position 371 (Δ371I), which disrupts the CD4-binding site[6]. We termed antibodies with this

profile "CH505 differential binders." We found that plasma from macaques immunized with TF or sequential CH505 Envs bound wild-type CH505 TF gp120 and equally well to the CD4-disrupted mutant protein. However, plasma from immunized macaques blocked binding of a CD4-binding site CH103 lineage member (CH106) and of sCD4 to CH505 TF gp120 Envs

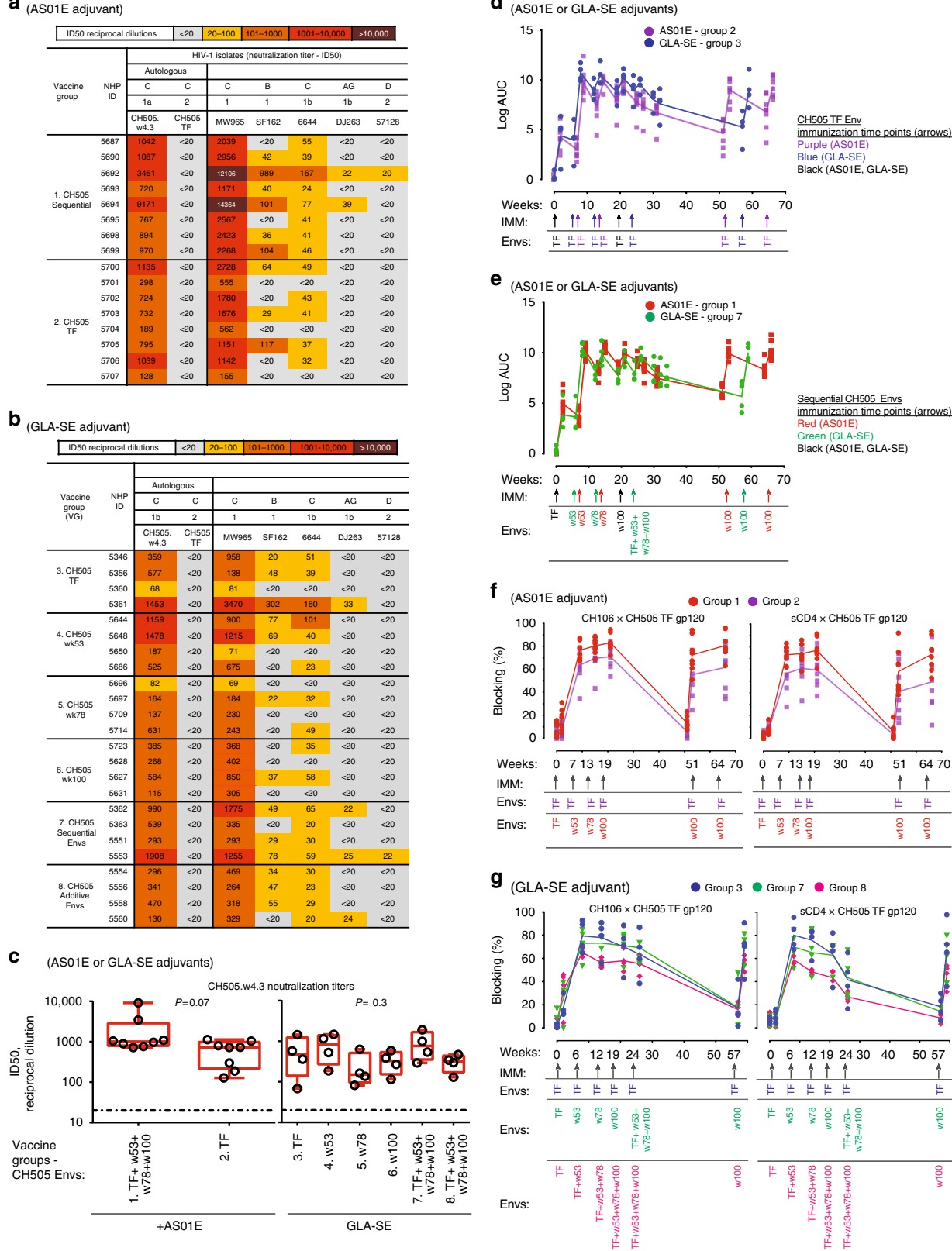

(Fig. 1f, g). These results further indicate that while there were high levels of CD4-binding site-like antibodies induced by CH505 Envs in macaques in plasma, this response did not have a predominance of differential-binding bnAb-like activity.

We extended our analysis of differential responses in CH505 Env-immunized macaques by enumerating the frequency of CH505 differential-binding memory (IgD[−], CD27 all) B cells induced at various time points (Fig. 2a; Supplementary Fig. 1), and found that macaques immunized with sequential CH505 gp120 Envs appeared to have an increased number of CH505 differential-binding memory B cells over background, compared to macaques immunized with CH505 TF gp120 Env alone. Furthermore, the number of responders for CH505 differential-binding memory B cells, and frequency of CH505 differential-binding memory B cells, appeared to go down after the third immunization, suggesting an off-target effect on CH505 differential-binding memory B cell lineages initiated by early CH505 Envs in the sequential Env regimens. To further examine CH505 differential-binding responses induced by TF-repetitive Env immunizations or sequential Env regimens in macaques, we recovered CH505 Env-reactive recombinant antibodies from sorted memory B cells isolated from animals immunized with TF Env in GLA-SE or AS01E, and compared their binding characteristics with those from macaques immunized with sequential combinations of Envs in GLA-SE or AS01E. One hundred and twenty antibodies were isolated using the TF Env gp120 as a fluorophore-labeled memory B cell hook from macaques immunized with TF Env (macaque $N = 5$), and 232 antibodies were isolated from macaques immunized with sequential combinations of Envs (macaque $N = 11$) (Fig. 2b, Table 1). Among macaques immunized with TF gp120 Env, 11% of the CH505 Env-reactive antibodies were CH505 differential binders; however, among those immunized with sequential combinations of CH505 gp120 Envs, 16% were CH505 differential binders (Fig. 2b). Thus, consistent with the observed trend of enhanced subdominant neutralizing antibody responses induced in the plasma by sequential immunization (Fig. 1a–c), we observed a trend for animals immunized with sequential CH505 gp120 Envs to make more CH505 differential-binding antibodies, albeit not statistically significant.

While the observed trends of enhanced plasma-neutralizing antibody responses, as well as higher frequencies of differential memory B cells and recombinant antibodies were seen in sequentially immunized macaques relative to those vaccinated with TF Env, no differences in the immunogenetics of CH505 differential-binding antibodies were observed between Ab groups

isolated from macaques immunized with TF alone in AS01E or GLA-SE, and sequential CH505 Envs in AS01E or GLA-SE (Fig. 2c, d). Strikingly however, among all CH505 Env-reactive antibodies in CH505 Env-immunized macaques (regardless of adjuvant), the heavy-chain genes of CH505 differential binders (macaque $N = 12$) were less mutated ($P = 0.008$, exact Wilcoxon test with false discovery rate (FDR) correction) and had longer HCDR3 lengths ($P = 0.008$, exact Wilcoxon test with false discovery rate (FDR) correction) than non-differential binder antibodies (macaque $N = 14$) (Fig. 2e, f). Non-HIV-1 antibodies (macaque $N = 13$) had similar mean heavy-chain gene-mutation frequencies, but shorter HCDR3 lengths than CH505 differential-binding antibodies. These data suggest that CH505 differential-binding antibodies arose from a pool of precursor B cells with longer HCDR3s and more limited affinity maturation capacity, features normally associated with B cells under clonal deletion/anergy host controls[16], and/or that are excluded from mature B2 subsets, and thus are predisposed to generating either suboptimal T-dependent responses or using T-independent pathways.

**Vaccine-induced macaque CD4-binding site-nAbs**. We selected 29 differential-binding recombinant antibodies based upon degree of antibody differential binding to wild-type and mutant CH505 Δ371I Envs, and autologous CH505.w4.3 virus neutralization or $V_H4$ usage (human CH103 used $V_H4$–59)[6] (Supplementary Fig. 2a, b). CH505 differential-binding antibodies isolated from macaques immunized with TF or sequential combinations of CH505 Envs had similar autologous Env-binding and neutralization patterns (Supplementary Fig. 2c, d). While CH505 differential-binding monoclonal antibodies (mAbs) neutralized autologous tier 1 CH505.w4.3 (Supplementary Fig. 2d), none neutralized the autologous tier 2 CH505 TF isolate. The antibody clonal lineage most similar to CH103 we identified was an IGHV4-J, CH505 differential-binding clonal lineage (DH522) with autologous tier 1, and heterologous tier 1 and weak tier 2 neutralization activities derived from memory B cells of RM-5556 immunized with additive CH505 Envs (Supplementary Fig. 2e). DH522 lineage antibodies, DH522.1 and DH522.2, were isolated from blood memory B cells after four and six immunizations, respectively (Fig. 3a). DH522 lineage mAbs did not neutralize tier 2 CH505 TF (Fig. 3b), neutralized 7% of a panel of 199 HIV-1 primary isolates (Supplementary Fig. 2e), and blocked the binding of CH103 lineage Ab (CH106) to CH505 TF gp120 (Fig. 3c). Chimeric Abs of heavy- and light-chain genes of DH522 and

---

**Fig. 1** Immunogenicity of CH505 Envelopes in macaques. Rhesus macaques were immunized with CH505 transmitted-founder (TF) or natural variants of TF (weeks (w) 53, 78, and 100) envelope (Env) alone, or sequential combinations of TF and TF[−] variants in AS01E or GLA-SE adjuvants. Immunization time points are indicated by an arrow. **a**, **b** Neutralization profiles of plasma post sixth immunization from macaques immunized with CH505 Envs in AS01E (**a**) or GLA-SE (**b**). Neutralization was assessed in TZM-bl cells once, except for titers <100 ID50 that are representative of two independent assays. **c** Post-sixth immunization plasma autologous neutralization titers. Each symbol represents the neutralization titer per animal, and the neutralization titers per group of animals are shown via box and whisker plots; box indicates median, and lower and upper quartile ranges, and whiskers indicate minimum and maximum titers. Neutralization positivity cutoff was 20 ID50 (dotted line). $P = 0.07$, exact Wilcoxon test; CH505.w4.3 neutralization titers for macaques immunized with CH505 TF alone or sequential Envs in AS01E. $P = 0.3$, Kruskal–Wallis test; CH505.w4.3 neutralization titers for macaques immunized with all CH505 Env regimens in GLA-SE. $P = 0.8$, exact Wilcoxon test; plasma neutralization titers for macaques immunized with CH505 TF Env in AS01E or GLA-SE. $P = 0.4$, exact Wilcoxon test; plasma neutralization titers from macaques immunized with sequential CH505 Envs in AS01E or GLA-SE. **d**, **e** Vaccination peaks and troughs for plasma reactivity with CH505 TF Env gp120; macaques were immunized with CH505 TF Env in AS01E or GLA-SE (**d**), and CH505 sequential Envs in AS01E or GLA-SE (**e**). Binding was performed in a single ELISA, each symbol represents plasma binding per animal, and the best fit line represents mean plasma binding within each group of animals. $P < 0.05$ is significant (exact Wilcoxon test); post sixth immunization, CH505 TF gp120 plasma-binding titers in macaques immunized with CH505 TF in AS01E ($9.3 \pm 1.0$) or GLA-SE ($8.9 \pm 1.6$) ($P = 0.8$) (**d**) and sequential CH505 Envs in AS01E ($10.2 \pm 0.9$) or GLA-SE ($10.0 \pm 0.8$) ($P = 0.7$) (**e**). **f**, **g** Plasma, post sixth immunization, competitive inhibition of soluble CD4 and CD4-binding site broadly neutralizing antibody (bnAb) CH106 in ELISA. Each symbol represents blocking levels per animal and the best fit line represents the mean blocking activity within each group of animals. Shown are representative plasma blocking activities of sCD4 and CH106 from macaques immunized with CH505 TF alone or sequential Env combinations (sequential, additive)

CH103 mAbs improved neither binding nor neutralization breadth of DH522 (Supplementary Fig. 2f), suggesting that compensatory mutations in CH103 heavy-and light-chain genes are necessary for mAb binding and neutralization.

It was previously reported that glycans occluded access to the CD4-binding site on SOSIP trimers by CD4-binding site bnAbs, including CH103[21, 22], and glycan-deleted trimers generally induced high titer autologous-nAbs in rhesus macaques or mice models[9, 22]. In the context of gp120 monomers, we found that the CH103 lineage antibodies bound equally well to natively glycosylated Envs and Envs with a deletion of glycans in the vicinity of the CD4-binding site[22] (Supplementary Fig. 3a).

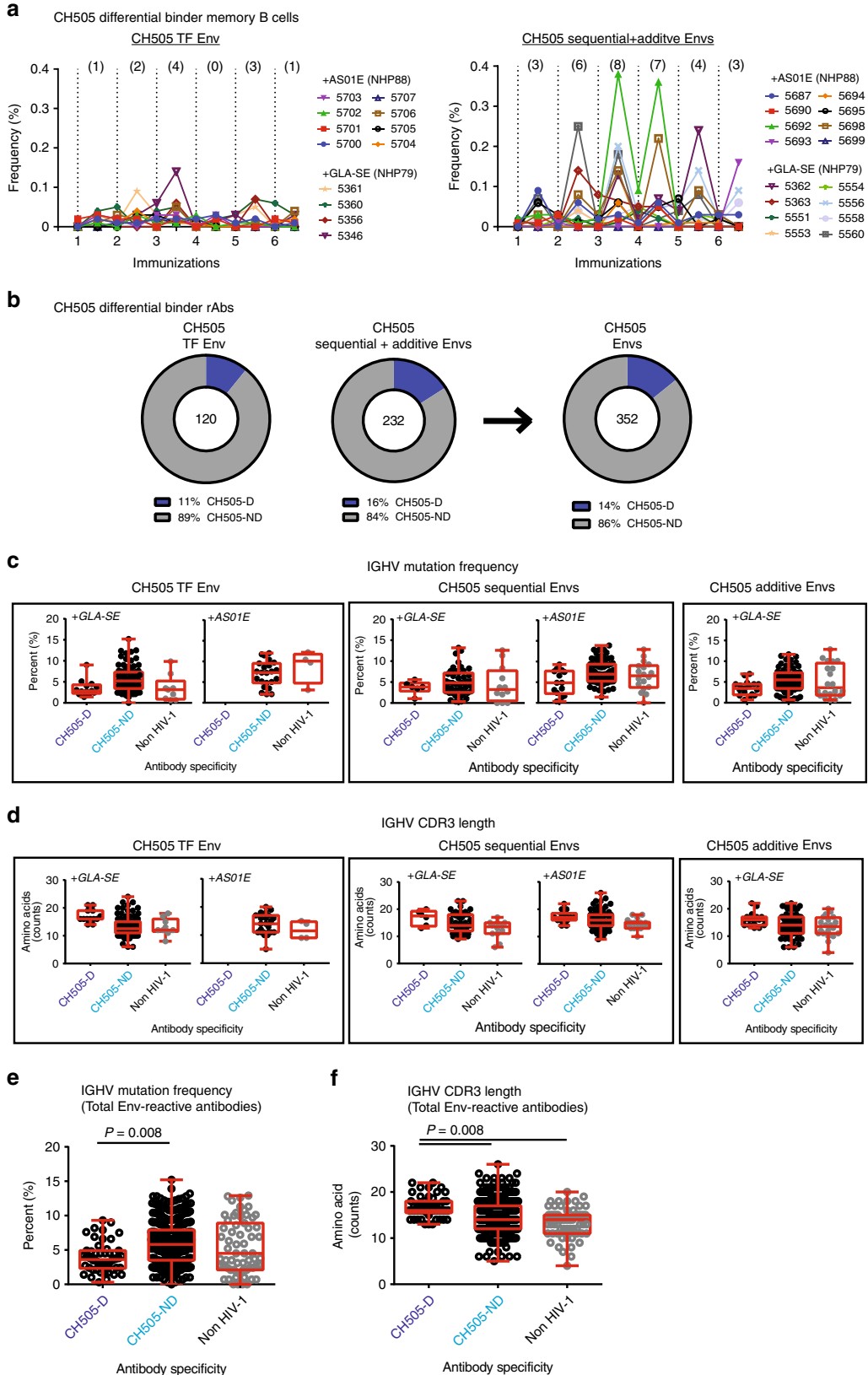

However, CH103 antibody demonstrated improved neutralization titers against HIV-1 strains bearing Envs with a deletion of glycans proximal to the CD4-binding site[22] compared to the viruses bearing natively glycosylated Envs (Supplementary Fig. 3b), suggesting that glycans in the vicinity of the CD4-binding site may impact the neutralization capacity of CH103 lineage antibodies. We also found that neither the UCA nor mature CH103 antibodies bound free N-glycans (Supplementary Fig. 3c). For comparison, we found that the DH522 lineage antibodies bound natively glycosylated and glycan-deleted CH505 TF gp120 Envs equally well, but did not neutralize HIV-1 pseudoviruses bearing these Envs nor bound free N-glycans (Supplementary Fig. 3). However, we also found DH522 lineage antibodies to be polyreactive with host antigens (Fig. 3d) as similarly reported for CH103 bnAb[6].

We determined the crystal structure of the DH522.2 Fab in complex with a deglycosylated chimeric B.YU2 gp120 core to 2.8 Å resolution (Supplementary Figs. 4 and 5 and Supplementary Table 1). We found that the DH522.2 HCDR3 was in an extended conformation, with its hydrophobic tip inserted into a pocket on the surface of gp120 near the junction of the inner domain, outer domain, and bridging sheet (Fig. 3e). This pocket was part of a broad hydrophobic surface underlaid by loop B and adjacent to the well-characterized Phe43 cavity[23–27]. A shift in the bridging sheet represented a departure from the CD4-bound state, exposing the hydrophobic patch rather than preserving the full depth and shape of the Phe43 cavity. The loop B hydrophobic pocket, the core epitope for DH522.2, is also the binding site of NBD-556 and related CD4 mimetic compounds (Fig. 3e, g)[28]. The HCDR3 hydrophobic tips of both DH522.2 and the poorly neutralizing CD4-binding site antibody F105 reach into the loop B pocket, but the overall orientations of the two Fabs with respect to the gp120 core are quite distinct (Fig. 3e). Although the DH522.2 HCDR3 probed the CD4-binding site Phe43 cavity, superposition of the DH522.2 Fab-gp120 complex onto the BG505 Env SOSIP.664 trimer indicated that DH522.2 Fab would clash with a nearby gp120 subunit (Fig. 3f). A 3D reconstruction of DH522.2-bound CH505 SOSIP.664 fully glycosylated trimer by negative stain electron microscopy confirmed that DH522.2-bound Env trimers in an open conformation (Fig. 3h–j), a mode of binding consistent with an Env conformation between the CD4- and b12-bound states (Fig. 3h). A broader comparison of CD4-binding site antibodies showed that other bnAbs included loop D and V5 in their epitopes rather than loop B (Fig. 3g); those bnAbs also had more favorable orientations of binding trimeric Envs with much less steric occlusion than found with DH522.2. Thus,

DH522.2 interacts poorly with a closed Env trimer, explaining the limited neutralization breadth of antibodies in its lineage.

**CH103 bnAb maturation in CH103 UCA $V_H DJ_H + V_L J_L$ KI mice.** While the rhesus IGHV4-J is similar to the human IGHV4–59 that encoded the CH103 bnAb heavy chain, an IGLV2 encoded the DH522 light chain, rather than an *IGLV3* gene as in CH103. One hypothesis for this difference is that receptor editing may have limited IGLV3 usage. A second hypothesis is that the macaque does not have a sufficiently similar IGLV3 that can pair with IGHV4-J and yield CH103 bnAb-like activity. Because the macaque Ig loci show high inter-individual diversity, we used next-generation sequencing (NGS) using IGVL3-specific primers to further probe macaque 5556 for germ line gene segments closer to IGLV3-1. The closest plausible germ line candidate is again 85% identical. Thus, these data suggested that insufficient macaque and human gene identity was a plausible reason for inability of macaques to make CH103-like bnAbs.

To study directly the immune mechanisms controlling development of HCDR3-binder type bnAbs and to more robustly reveal potential immunogenicity differences between vaccination groups in a less stringent (i.e., less competitive polyclonal system), we turned instead to a humanized immunoglobulin mouse model. We generated a double KI mouse, designed to express the rearranged CH103 UCA heavy- and light-chains (double KI; $V_H DJ_H{}^{+/+} + V_L J_L{}^{+/-}$). Naive B cells from naive CH103 UCA double KI mice were not clonally deleted at the first (central) tolerance checkpoint, that is, at the pre-B to immature B cell transition in BM (Fig. 4a) as observed in naive 2F5 germ line/UA double KI mice[16] (Supplementary Fig. 6a). Instead, they exhibited a blockade at the BM transitional to mature B cell stage (70% mean reduction relative to WT) (Fig. 4a, b), and a subpopulation of residual mature B cells with lowered B cell receptor (BCR) densities (Fig. 4a). In line with specific negative selection of mature BM B cells, CH103 UCA double KI mice also had lower numbers of total peripheral B cells (Fig. 4c, d), a larger fraction of which accumulated in the splenic transitional compartment (Fig. 4c, e). Thus, these data are consistent with the processes of both anergy and deletion of CH103 UCA double KI-expressing positive B cells at the second tolerance checkpoint.

**Light-chain editing in CH103 UCA KI mice.** Since self-reactivity of bnAbs often correlate with their HIV-1 Env reactivity[18, 29], we determined the frequency of CH505 differential-binding (CH505 TF gp120+, CH505 TF gp120 Δ371I–.) B cells in the mature

**Fig. 2** Repertoire analysis of CH505 Envelope vaccine-induced antibodies. Memory B cells bearing candidate CD4-binding site receptors, or recombinant CD4-binding site antibodies, are termed CH505 differential binders. **a** Frequency of CH505 differential-binding memory B cells from 12 macaques immunized with CH505 TF envelope (Env) gp120 alone and 16 macaques immunized with sequential combinations of CH505 Env gp120s (sequential + additive). One million peripheral blood cells (PBMCs) from immunized macaques were phenotyped by FACS analysis for memory B cells that had CH505 Env differential binding. Shown is frequency (%) of memory B cells in PBMC that demonstrated CH505 Env differential binding. Number of animals with CH505 differential-binding memory B cells above background at 2 weeks post each immunization is listed above each graph. **b** Total Env-reactive antibodies isolated from blood memory collected 2 weeks post fourth and sixth immunizations in 16 macaques immunized with CH505 TF alone or sequential combinations of Env gp120, regardless of adjuvant; CH505 differential (CH505-D) and non-differential binders (CH505-ND). **c, d** Immunogenetics of antibodies isolated from macaques who received similar CH505 Envs, but different adjuvant. Antibody heavy variable (IGHV) gene mutation frequencies (**c**) and CDR3 lengths (**d**) were inferred by Cloanalyst software program and the results shown via box and whiskers plots; box indicates median, and lower and upper quartile ranges, and whiskers indicate minimum and maximum numbers evaluated. **e** IGHV mutation frequencies, and (**f**) IGHV CDR3 lengths, of total Env-reactive antibodies isolated from macaques regardless of CH505 Env vaccines or adjuvants. **c–f** Statistical analyses: (**c, d**) $P > 0.05$ (exact Wilcoxon test with false discovery rate (FDR) correction)—(i) IGHV mutation frequencies or CDR3 lengths of CH505 differentials or non-differentials isolated from animals receiving similar Env vaccines (TF or sequential Envs) in different adjuvants, and (ii) immunogenetics of CH505-D and CH505-ND antibodies isolated from macaques immunized with sequential or additive CH505 Envs in GLA-SE; (**e, f**) $P = 0.008$ (exact Wilcoxon test with FDR correction)—IGHV mutation frequency and CDR3 lengths for CH505-D and CH505-ND antibodies, regardless of Env vaccines and adjuvants, and (ii) for IGHV CDR3 lengths of CH505-D and non HIV-1-reactive antibodies

**Table 1 Frequency of antibodies isolated from CH505 envelope-vaccinated macaques**

| CH505 vaccine group | Animal ID | All antibody lineages isolated | | | | | Unique clonal lineages[a] | | | | |
|---|---|---|---|---|---|---|---|---|---|---|---|
| | | Total Abs | Non HIV-1 | CH505 Env$^+$ | CH505 differentials | | Total unique Abs | Non HIV-1 | CH505 Env$^+$ | CH505 differentials | |
| | | | | | Count | % Of total Abs | | | | Count | % Of total Abs |
| CH505 TF alone | 5346 | 15 | 2 | 13 | 4 | 27% | 13 | 2 | 11 | 2 | 15% |
| | 5356 | 2 | 1 | 1 | 0 | 0% | 2 | 1 | 1 | 0 | 0% |
| | 5360 | 11 | 2 | 9 | 0 | 0% | 10 | 2 | 8 | 0 | 0% |
| | 5361 | 78 | 4 | 74 | 9 | 12% | 68 | 4 | 64 | 7 | 10% |
| | 5703 | 27 | 4 | 23 | 0 | 0% | 25 | 4 | 21 | 0 | 0% |
| CH505 sequential | 5362 | 2 | 0 | 2 | 1 | 50% | 2 | 0 | 2 | 1 | 50% |
| | 5363 | 14 | 0 | 14 | 1 | 7% | 14 | 0 | 14 | 1 | 7% |
| | 5551 | 11 | 1 | 10 | 1 | 9% | 11 | 0 | 10 | 1 | 9% |
| | 5553 | 35 | 11 | 24 | 3 | 9% | 34 | 11 | 23 | 3 | 9% |
| | 5692 | 40 | 5 | 35 | 4 | 10% | 39 | 5 | 34 | 4 | 10% |
| | 5694 | 34 | 12 | 22 | 6 | 18% | 33 | 12 | 21 | 6 | 18% |
| | 5699 | 36 | 1 | 35 | 4 | 11% | 29 | 1 | 28 | 4 | 14% |
| CH505 additive (sequential combination) | 5554 | 23 | 4 | 19 | 3 | 13% | 21 | 4 | 17 | 3 | 14% |
| | 5556 | 49 | 10 | 39 | 9 | 18% | 43 | 10 | 33 | 7 | 16% |
| | 5558 | 28 | 5 | 23 | 4 | 14% | 25 | 4 | 21 | 4 | 16% |
| | 5560 | 10 | 1 | 9 | 1 | 10% | 10 | 1 | 9 | 1 | 10% |
| Summary (total/%) | | 415 | 63 | 352 | 50 | 12% (50/415) | 379 | 61 | 317 | 44 | 12% (44/379) |

Abs, antibodies; Env, envelope
We isolated 415 total antibodies from 16 adult rhesus macaques vaccinated with CH505 TF alone or sequential envelope regimens (sequential, additive). Antibodies were isolated post fourth immunization in all macaques as well as post sixth immunization in animals 5361 and 5556. Here, we report the number of antibodies isolated from each macaque per vaccine group studied, and the frequency of CH505 differential binders. CH505 differential-binding antibodies (CH505 TF gp120+, CH505 TF gp120 Δ371I+/− (greater than or equal to threefold vs. WT)) were referred to as CH505 differentials.
[a] A singleton or ≥2 antibodies with the same VH and JH rearrangement, and HCDR3 length, paired with the same light chain were counted as a single unique antibody. Antibody immunogenetics were inferred by Cloanalyst software program

follicular subset of naive CH103 UCA double KI mice relative to those that accumulated at the second tolerance checkpoint blockade, that is, in the transitional subset. Indeed, the majority of B cells in the follicular mature pre-immune subset lost CH505 differential reactivity, whereas splenic transitional B cells were significantly enriched for TF Env binding (Fig. 5a), with up to 68% exhibiting CH505 differential reactivity (Fig. 5b). Moreover, the percentages of peripheral mature B cells that retained their differential TF Env specificity varied among individual mice, and this variance inversely correlated with positive B cell selection into the mature subset (Fig. 5c) (Pearson $R$ score = −0.85; $P$ = 0.004, Student's $t$ test).

That CH103 UCA double KI mice had residual mature B cells with modified BCR specificities (lacking differential CH505 Env binding), combined with previous findings that the CH103 LC was required for differential binding[6], suggested that the KI UCA LC contributed to tolerization of CH103UCA double KI B cells. To test this possibility, we compared CH103 UCA double KI mice with $V_HDJ_H^{+/+}$ KI mice, which can only use endogenous mouse LCs. We found that CH103 UCA $V_HDJ_H^{+/+}$ mice had normal B cell development in BM (Fig. 5d; Supplementary Fig. 6b) as well as typical total B cell numbers (Supplementary Fig. 6c) and mature B cell distributions in the periphery (Supplementary Fig. 6d), thus rescuing the B cell block seen in CH103 UCA double KI mice. These results indicated that the UCA LC of CH103 contributed to its negative selection in vivo and predicted that residual mature B cells in CH103 UCA double KI mice should be substantially enriched for receptor-edited KI LCs, as in the low-affinity 3H9 α-DNA KI mouse model[30].

To evaluate formally if receptor editing of the CH103 UCA-expressing B cells was occurring in vivo, we generated a heterozygous version of the CH103 double KI model (het; $V_HDJ_H^{+/−}$ + $V_LJ_L^{+/−}$). We combined heterozygous expression of the KI LC allele with a specific targeting strategy, in which we knocked in the human CH103 $V_\lambda J_\lambda$ rearrangement at mouse Igκ

(rather than Ig$_\lambda$; Supplementary Fig. 7) and retained downstream Jκ4/5 mini-gene segments[16, 18]. This approach allowed CH103 UCA het double KI mice to undergo multiple secondary LC rearrangements both in cis and in trans, and generated a more diverse repertoire of VJ rearrangements than possible at the atypically abbreviated mouse $_\lambda$ locus.

As predicted by their potential for both alternative κLC and HC rearrangement, B cell subsets from naive CH103 UCA heterozygous double KI spleen had significantly lower CH505 differential reactivity than did those from the CH103 UCA double KI model, including a fourfold reduction in differential binders found in their transitional B cell subset, and proportionally even lower differential reactivity in the mature B cell subset (more than sevenfold reduction in differential binders) (Fig. 5e). The result was a near complete lack of Env reactivity, that is, >98% of mature B cells. As expected, loss of Env reactivity in CH103 UCA heterozygous double KI mice was also associated with rescue of mature B cells development, which resembled the normal developmental phenotype of homozygous CH103 UCA VDJ$^{+/+}$ KI mice (Supplementary Fig. 6a–d).

We also isolated by RT-PCR $V_HDJ_H$ and $V_\lambda J_\lambda$ Ig gene pairs from single splenic mature follicular B cells in naive CH103 het double KI mice. Half (23/46) of the sorted mature B cells used the KI $V_HDJ_H$ rearrangement, and only 1/23 (4.3%) of the KI $V_HDJ_H$-expressing clones used the original CH103 UCA $V_\lambda$-J$_\lambda$1 rearrangement (Fig. 5f), instead expressing Vκ5-39 almost exclusively. In addition, single-sorted clones were skewed toward Jκ4/5 usage (Fig. 5f), further indicating selection for κ-edited clones and suggesting that the CH103 UCA $V_\lambda J_\lambda$ rearrangement had been replaced by secondary rearrangement events on the KI allele. Moreover, we did not detect by flow staining higher $_\lambda$LC usage in mature B cells from CH103 het double KI mice than in similar cells from wild-type (WT) B6 mice (Supplementary Fig. 6e), nor did we detect by single-cell repertoire analysis any single-cell clones bearing $V_\lambda J_\lambda$ rearrangements (Fig. 5f).

To examine if the CH103 UCA had poly and/or autoreactivity in vitro, and if LC editing of the KI LC using the editor Vκ5-39 could eliminate or mitigate it, we performed the >9600 host proto-array screen[29] on the CH103 UCA mAb and a representative recombinant mAb, M5808-4D1, consisting of the natural $V_H DJ_H/V_\lambda J_\lambda$ pair from a single mature B cells (sorted from a CH103 UCA het double KI mouse) expressing an unmutated CH103 UCA HC, paired with a LC bearing the editor Vκ5-39-Jκ5 rearrangement. The CH103 UCA mAb had minimal polyreactivity but had moderate affinity for several host protein and high

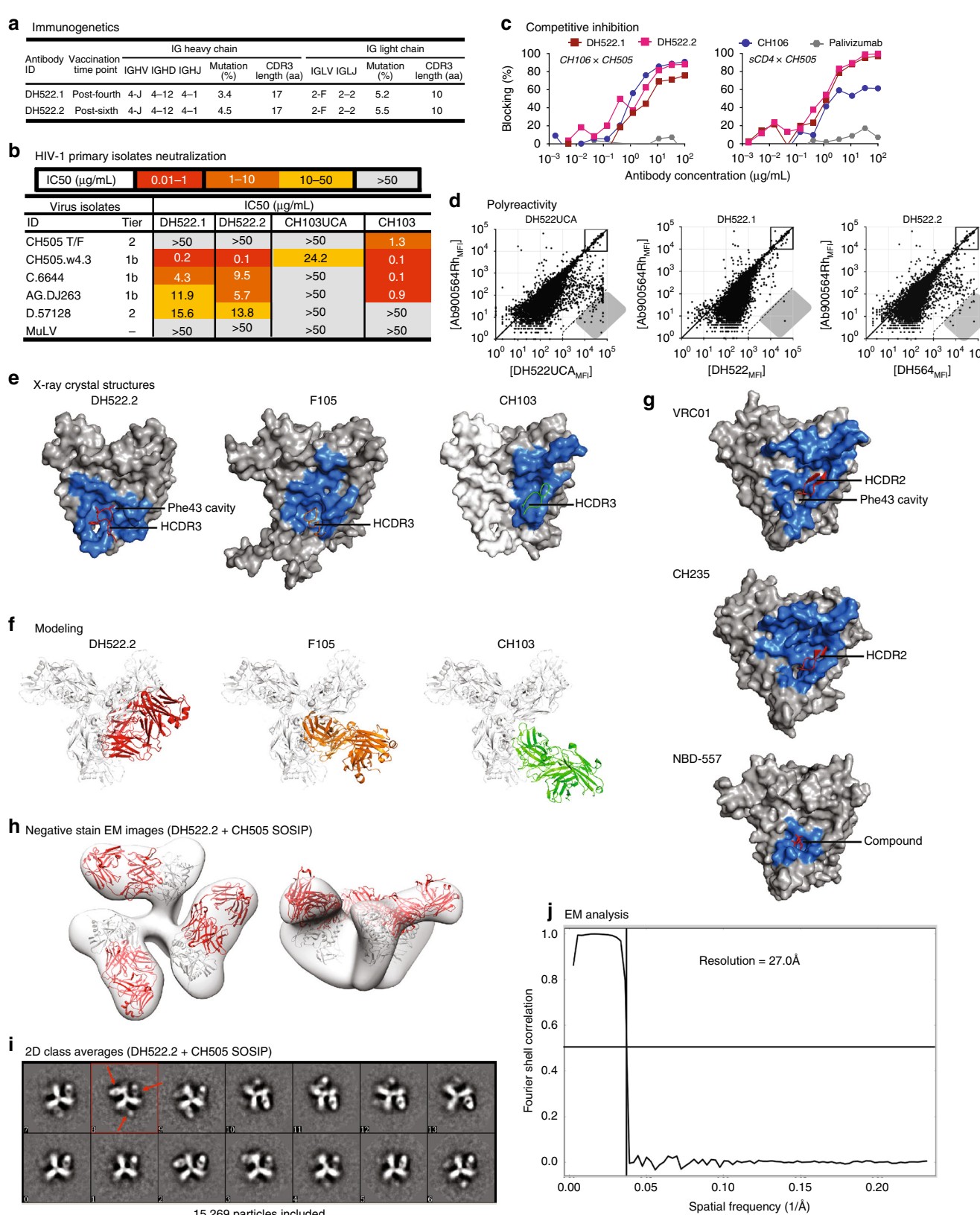

**a** Immunogenetics

| Antibody ID | Vaccination time point | IG heavy chain | | | | | IG light chain | | | |
|---|---|---|---|---|---|---|---|---|---|---|
| | | IGHV | IGHD | IGHJ | Mutation (%) | CDR3 length (aa) | IGLV | IGLJ | Mutation (%) | CDR3 length (aa) |
| DH522.1 | Post-fourth | 4-J | 4–12 | 4–1 | 3.4 | 17 | 2-F | 2–2 | 5.2 | 10 |
| DH522.2 | Post-sixth | 4-J | 4–12 | 4–1 | 4.5 | 17 | 2-F | 2–2 | 5.5 | 10 |

**b** HIV-1 primary isolates neutralization

| IC50 (µg/mL) | 0.01–1 | 1–10 | 10–50 | >50 |
|---|---|---|---|---|

| Virus isolates | | IC50 (µg/mL) | | | |
|---|---|---|---|---|---|
| ID | Tier | DH522.1 | DH522.2 | CH103UCA | CH103 |
| CH505 T/F | 2 | >50 | >50 | >50 | 1.3 |
| CH505.w4.3 | 1b | 0.2 | 0.1 | 24.2 | 0.1 |
| C.6644 | 1b | 4.3 | 9.5 | >50 | 0.1 |
| AG.DJ263 | 1b | 11.9 | 5.7 | >50 | 0.9 |
| D.57128 | 2 | 15.6 | 13.8 | >50 | >50 |
| MuLV | – | >50 | >50 | >50 | >50 |

**c** Competitive inhibition

**d** Polyreactivity

**e** X-ray crystal structures

**f** Modeling

**g** VRC01 / CH235 / NBD-557

**h** Negative stain EM images (DH522.2 + CH505 SOSIP)

**i** 2D class averages (DH522.2 + CH505 SOSIP)

15,269 particles included

**j** EM analysis

Resolution = 27.0Å

affinity for two, BAG5 and MCCC1 (Fig. 5g; left). The $V_L$-edited M5808-4D1 Ab did not bind these two candidate autoantigens, and it also lacked the moderate reactivity with various host proteins characteristic of the CH103 UCA (Fig. 5g). Thus, most naive mature B cells with Vκ5-39 had low autoreactivity but also no binding to CH505 Env.

**BCR responsiveness in naive CH103 UCA double KI mice.** To study Env-reactive and Env-non-reactive mature follicular subpopulations, we used CH103 UCA double KI (homozygous) mice with sufficiently high frequencies of CH505 differentially binding double KI-positive ("Env+") clones (~10%; Fig. 5b, e) to allow their detection for comparison with LC-edited ("Env−") clones. Since one of the hallmarks of anergic B cells is downmodulation of surface IgM[31], we used flow cytometry to measure IgM mean fluorescence intensities (MFIs) in both transitional and mature B cell subsets, further subfractionated based on Env reactivities. Both Env+ and Env− transitional B cell subsets had IgM MFIs comparable to those in WT B6 mice. In contrast, Env+ mature B cells from naive CH103 UCA double KI mice had lower surface IgM densities than did Env− mature B cells (Fig. 6a, b), which had IgM MFIs comparable to those of WT B6 transitional B cells (Fig. 6b), suggesting that the Env+ cells had become anergic after bypassing the second checkpoint. The majority of mature B cells that had undergone LC editing (and lost Env reactivity) were not only rescued from clonal deletion but also rescued from anergy.

Consistent with the above IgM MFI data, both Env+ and Env− transitional B cells from CH103 UCA double KI mice had Ca++ responses to saturating concentrations of anti-Ig similar to those of transitional B cells from WT (B6) mice (Fig. 6c, d). At sub-saturating concentrations, however, Env+ splenic transitional cells had attenuated Ca++ responses (Fig. 6d), suggesting they were partially anergic. In contrast, Env+ mature B cells from CH103 UCA double KI mice had modestly attenuated Ca++ responses to saturating concentrations of anti-Ig, but more attenuated Ca++ signaling at limiting amounts of anti-Ig (Fig. 6c, d). This gradient of anergy, in which the most accentuated decreases in BCR signaling capacity were in mature Env+B cell subsets, either present in mature follicles or recirculating in BM, rather than in the prototypical T3 transitional compartment[32], is analogous to auto-Ab KI models in which anergic clones have also been found in various mature B cell subsets[33, 34].

**Env activation of CH103 double KI-positive B cells.** TF Env gp120 binds the CH103 UCA IgG antibody in vitro more tightly ($K_D$ in the high nM range[6]) than activation thresholds estimated for triggering naive B cells in vivo[35]. Given the anergic status of naive CH103 UCA double KI-positive mature B cells (Fig. 6a–d), however, such clones could require stronger BCR and/or T cell signals, the amounts of which can vary depending on stage and degree of silencing[32].

To test whether TF Env could activate the naive B cell repertoire in CH103 UCA double KI mice, we examined Ca++ responses of their transitional and mature B cells ex vivo, using TF Env concentrations that saturated their BCRs (Fig. 6e). Since we could not directly compare responses in Env+ and Env− subsets (due to cross-blocking issues associated with using TF Envs both as a detection and BCR-stimulating reagent), we evaluated total transitional and mature subsets that are enriched for Env+ and Env− clones, respectively. TF Env gp120 tetramers induced Ca+ flux near levels induced by the anti-Ig control in transitional B cells from CH103 UCA double KI mice, but not in WT B6 mice (Fig. 6e; bottom left), showing that double KI clones could be specifically activated by this CH103 UCA-targeting immunogen. The effect was not seen with TF gp120 monomers (Supplementary Fig. 8a), indicating a requirement for multimeric presentation of antigens to mediate the Ca++ flux in anergic double KI-positive clones. Similar results were also seen in peripheral (splenic) transitional and mature B cells (Supplementary Fig. 8b).

TF Env gp120 tetramers induced low levels of Ca++ mobilization (10% of those induced by the anti-Ig control) in mature B cell subsets of CH103 UCA double KI mice (Fig. 6e; bottom right), but did not induce Ca++ flux in mature B cells of WT B6 mice (Fig. 6e; bottom row). While the low, but detectable Ca++ induction may have been expected, given the anergic status of the double KI mature B cell subset, the level induced was similar to the percentage of clones in the subset that retained Env reactivity (10%). Thus, despite their anergic status, these mature CH103 UCA expressing B cell clones can respond to TF Env gp120, provided that the latter is present at high enough concentration and in multimeric form.

**Vaccine-induced responses in CH103 UCA double KI mice.** We next studied B cell responses from CH103 het double KI mice immunized either twice with TF Env in the TLR4 agonist adjuvant, GLA-SE, or after the first two steps of the 4-valent sequential vaccine regimen, that is, TF and week 53 CH505 Envs[6] (both in GLA-SE), to evaluate whether immunization could initiate the CH103 bnAb B cell lineage (Fig. 7a, b, Supplementary

**Fig. 3** Characteristics of macaque DH522 lineage antibodies. **a** Immunogenetics of clonally related DH522 lineage antibodies inferred by the Cloanalyst software program. DH522.1 and DH522.2 antibodies were isolated after 4 (21 weeks) and 6 (59 weeks) immunizations, respectively. **b** Neutralization profile of DH522.1 and DH522.2 monoclonal antibodies (mAbs) for viruses previously tested by plasma from CH505 Env-vaccinated macaques. Neutralization was performed in TZM-bl cells and titers reported as μg/mL IC50. **c** DH522.1 and DH522.2 competition for binding epitopes targeted by CD4-binding site bnAb CH106 and soluble CD4. Palivizumab was used as a negative control. **d** Binding profile of DH522 lineage antibodies to a panel of >9600 autoantigens via luminex as previously described[44]. Shaded gray region; autoantigen binding to each mAb, and a polyreactivity profile for each mAb. **e** Footprints of DH522.2 and two related antibodies, on the surface of deglycosylated gp120 Env monomeric cores. Surface of deglycosylated gp120 in gray and antibody footprint in blue; the surface of deglycosylated HIV-1 YU2 gp120 core in the DH522.2 complex structure is underlain by a trace of the deglycosylated gp120 core in the compound-probed structure. (DH522.2) Structure of DH522.2—hydrophobic side chains of the Val-Leu-Phe motif at the tip of DH522.2 HCDR3 interact with a pocket on the surface of gp120 lined by loop B and adjacent to the CD4-binding site; (F105) the Val-Phe-Tyr hydrophobic tip of the F105 HCDR3 interacts with the same hydrophobic pocket on gp120 (PDB: 3HI1)[45]; (CH103) the CH103 footprint rests heavily on the CD4-binding loop and V5 (PDB: 4JAN)[6]. Crystal structure of the CH103 complex had only the gp120 outer domain; the inner domain (lighter gray) was modeled from the DH522.2-bound structure. **f** Superposition onto the BG505 SOSIP.664 trimer model of the deglycosylated gp120 cores in structures of complexes with DH522.2, F105, and CH103; view is along the gp140 threefold axis. **g** Footprints of CD4-mimic, CD4-binding site bnAbs (VRC01, CH235), and CD4 attachment inhibitor *N*-(4-bromophenyl)-*N*′-(2,2,6,6-tetramethylpiperidin-4-yl)ethanediamine (NBD-557)[28]. **h–j** Negative stain electron microscopy analysis of DH522.2 in complex with the fully glycosylated CH505 transmitted-founder (TF) SOSIP.664 trimer. **h** Electron microscopy (EM) reconstruction of a DH522.2-SOSIP.664 complex; b12-bound gp120 was docked into the EM map (PDB: 2NY7)[46]. **i** 2D class averages of the DH522.2-SOSIP.664 complex. Fabs indicated by the red arrows. **j** The Fourier shell correlation curve is shown along with the resolution and is determined using FSC = 0.5

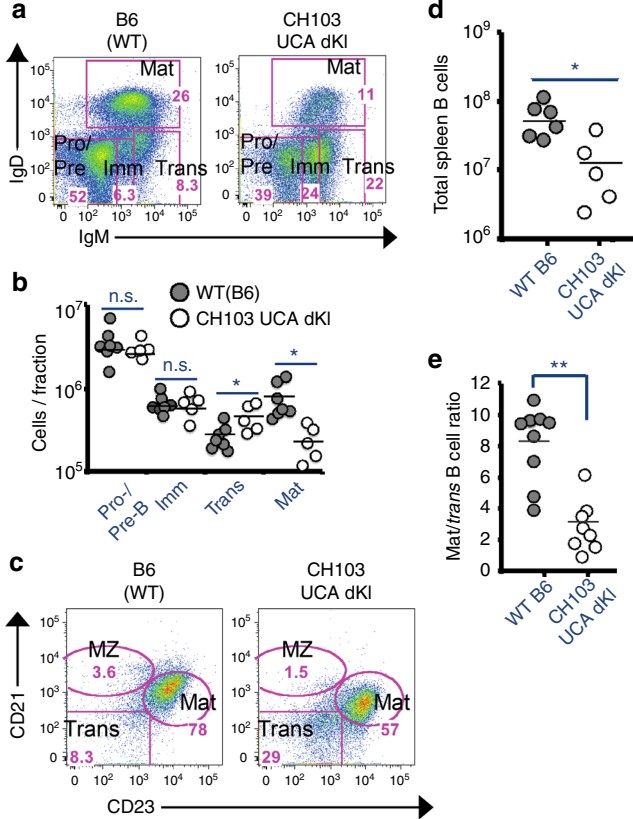

**Fig. 4** Immune tolerance in CH103 germ line knock-in mice. **a–c** Bone Marrow B cell development in naive CH103 unmutated common ancestor (UCA) double knock-in (dKI) mice, compared to wild type (B6), showing anergic phenotype and reduced frequencies of mature B cells associated with the second tolerance checkpoint. **a** Flow histograms indicating percentages in progenitor/precursor (Pro/pre), Immature (Imm), Transitional (Trans), and Mature (Mat) subsets. Data gated on live, total (CD19+ B220+) B cells. **b** Graphical summary of data shown in (**a**) for several mice. **c–e** Peripheral B cell development in naive CH103 UCA dKI mice, showing reduced total B cell numbers and frequencies of follicular mature B cells. **c** Representative flow histograms indicating percentages of total, live gated splenic B cells in transitional (Trans), marginal zone (MZ), and mature follicular (Mat) subsets. Graphical representation of total splenic B cell numbers (**d**) or mature to transitional ratios (**e**), as a measure of developmental arrest (transitional B cell accumulation) in periphery. Circles represent individual mice, and means are denoted by black bars. *P < 0.05; **P < 0.005, two-tailed Student's t-test. n.s. not significant

Fig. 9). Relative to control-immunized (saline-injected) mice and adjuvant-only immunized (GLA-SE administered) mice, detectable differential-binding IgG+ memory B cell subsets were elicited by both immunization regimens, but sequential immunization with TF and week 53 Envs induced >10-fold higher absolute numbers of differential-binding IgG+ splenic memory B cells than did immunization twice with TF Env (Fig. 7a, b). Higher frequencies of differential-binding, switched memory B cells were also seen in draining lymph nodes of TF and week 53 gp120 Env sequentially immunized mice than in nodes of mice immunized twice with the TF Env alone (Supplementary Fig. 9a). Some IgM+ differential binders could be expanded by GLA-SE alone (Supplementary Fig. 9c), indicating that TLR4 signaling can activate anergic CH103 double KI clones in vivo but alone can neither induce their class switch nor memory. Env immunizations also induced a second Env-specific IgG+ memory B cell population: non-differential Env binders, representing ~35% of clones within the TF Env-specific IgG+ memory pool of sequential Env-immunized animals, as calculated by subtracting the TF Env "differential-binding" IgG+ pool (Fig. 7a; bottom right) from the total TF Env-binding IgG+ pool (Fig. 7a; top right). This additional, non-differential IgG+ memory B cell population induced by Env immunization likely represents non-anergic clones with cross-reactivity to Env gp120 "off-target" epitopes, by virtue of using either endogenous (non-CH103) HCs and/or non-CH103

nor Vκ5-39 LCs. As a result of being preferentially selected, these B cells already begin to be amplified (above their sub-detection levels in the naive repertoire) at this earlier immunization phase.

Consistent with the flow cytometry data, both immunization regimens induced robust Env-specific serum IgG titers, whereas control (that is, saline or adjuvant only) injections failed to elicit detectable TF Env+ serum IgG titers (Fig. 7c). Repeated TF Env immunization predominantly induced non-differential serum IgGs, while sequential administration (with TF and week 53 Env) preferentially induced low and transient, but detectable titers of differential-binding serum IgG responses (Fig. 7c), suggesting that early phases of the 4-valent sequential vaccine regimen partially overcomes anergy controls of differential-binding, unedited (double KI-expressing) B cells. Furthermore, serum Ab responses elicited in sequentially immunized mice (but not repeatedly immunized with TF Env) developed low, but detectable, neutralizing titers against the autologous tier 1 virus CH505.w4.3 as listed in Supplementary Table 2, similar to neutralizing properties of the CH103 UCA[7]. Thus, analogous to the enhanced serum nAb responses (Fig. 1a–c) and increased frequencies of differential-binding (bnAb-like) memory B cells and recombinant antibodies (Fig. 2a, b) observed in sequentially immunized macaques (the latter which have characteristics consistent with being self-reactive; Fig. 2f), our results here demonstrate that the minor subset of anergic, lineage-specific

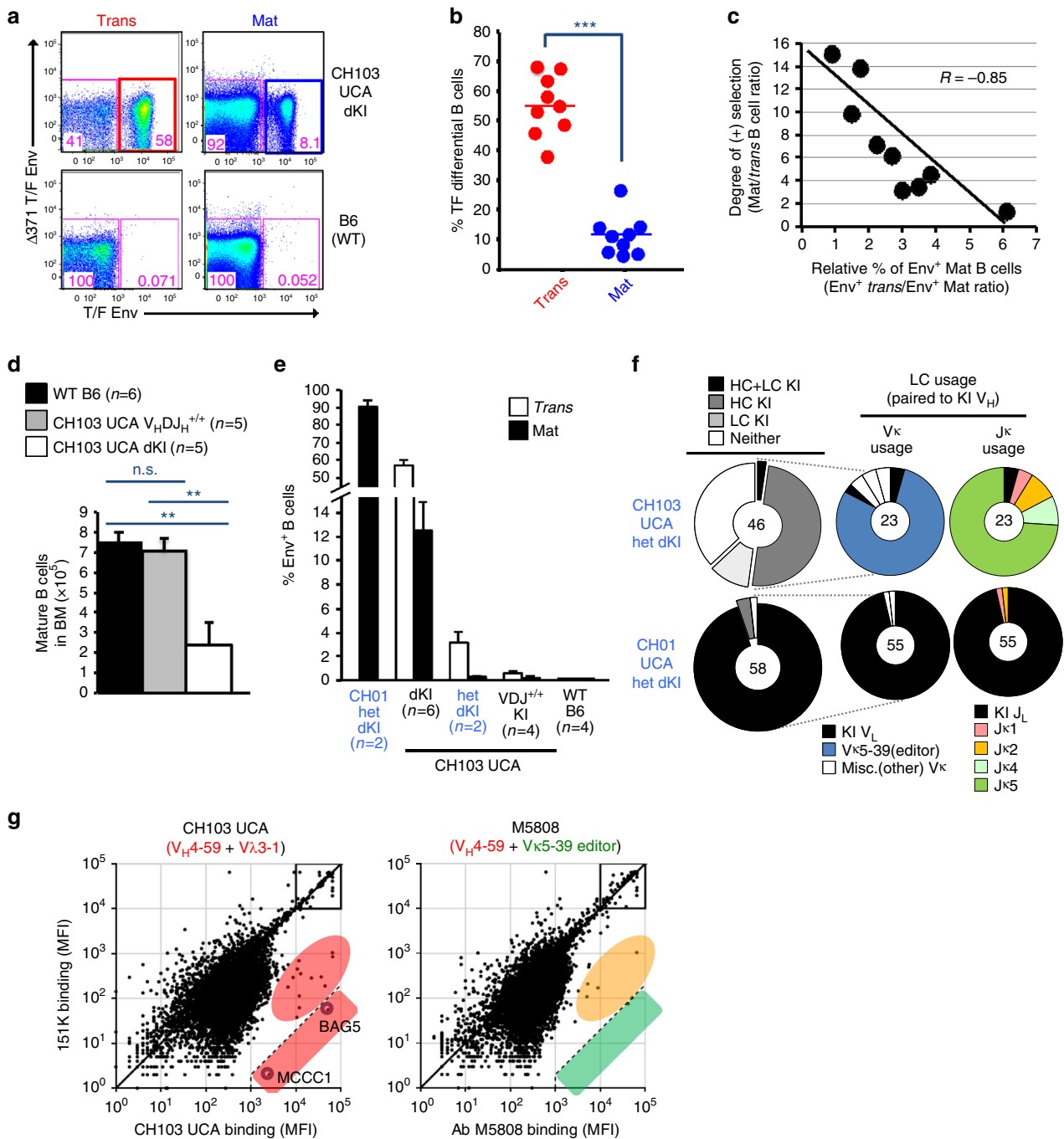

**Fig. 5** Analysis of receptor editing in CH103 germ line knock-in mice. **a–c** Env reactivity in mature B cell compartments of CH103 unmutated common ancestor (UCA) double knock-in (dKI) mice. **a** Representative pseudocolored dot plots showing typical patterns of reduced TF Env⁺ mature (follicular) B cells in naive CH103 UCA dKI mice. Shown as controls for background TF Env staining are WT (B6) mice. **b** Graphical summaries of percentages of differential binding in transitional and mature compartments in naive CH103 UCA dKI mice. Each dot represents an individual animal. ***$P < 0.001$, two-tailed Student's $t$ test. **c** Regression analysis of developmental blockade severity and residual mature B cell fraction retaining Env specificity. **d** Loss of CH103 KI light chain (LC) rescues splenic mature B cell development and reverses anergic phenotype in CH103 UCA $V_HDJ_H^{+/+}$ ("heavy-chain (HC)-only") KI mice. **$P < 0.005$, two-tailed Student's $t$ test. **e** Progressively decreased differential binding of splenic transitional and mature B cells (means + SEM) in CH103 UCA dKI ($n = 6$), heterozygous (het) dKI ($n = 5$), and $V_HDJ_H^{+/+}$ KI ($n = 4$) mice. CH01 het dKI mice ($n = 2$), a model that undergoes no negative selection, and WT B6 mice ($n = 4$) are shown as positive and negative controls for Env binding, respectively. **f, g** Extensive and highly-restricted receptor editing of the CH103 $\lambda$3-20 LC by "master editor" V$\kappa$5-39, as revealed in CH103 UCA het dKI mice. **f** Receptor editing in CH103 UCA het dKI mice. Shown are pie charts of HC/LC usage and breakdown of V$\kappa$ family and J$\kappa$ usage, among LCs paired to the KI $V_H$4-59-bearing HC. Individual mature B cells from naive CH103 het dKI mice were obtained by single-cell sorting on the total (unselected for TF Env binding) mature follicular B cell repertoire, and $V_HDJ_H$/$V_\lambda J_\lambda$ Ig gene pairs from single cells were recovered by RT-PCR for sequencing and immunogenetic analysis. **g** Polyreactivity/autoreactivity profiles of the CH103 UCA and V$\kappa$5-39 editor M5808 monoclonal antibodies (mAbs) on human protoarrays. The >500-fold binding compared to control mAb, that is, the official autoreactivity "cutoff", is indicated by dashed lines. Also shown are candidate autoantigens bound by the CH103 UCA mAb (circled in blue, in the red shaded box) and to which binding by M5808 is eliminated (green-shaded box)

precursors in follicles of CH103 UCA heterozygous double KI mice can be initially induced to expand, switch, secrete serum Abs, and form antigen-specific memory in vivo, in response to sequential CH103 lineage-targeting Env immunogens.

We sorted single, differential-binding IgG⁺ memory splenic B cells after the second immunizations and recovered HC/LC pairs for sequence analysis. As expected, differentially sorted IgG⁺ memory B cells from immunized mice had higher frequencies of KI HC-expressing clones that paired with KI LCs than did total IgG⁺ memory B cells from control mice immunized with saline or GLA-SE alone (Fig. 7d, e). This enriched, immunization-induced, IgG⁺ differentially sorted population (32% overall, compared to 4% in IgG⁺B cells from saline-immunized animals), confirms that some of the differentially binding memory population in

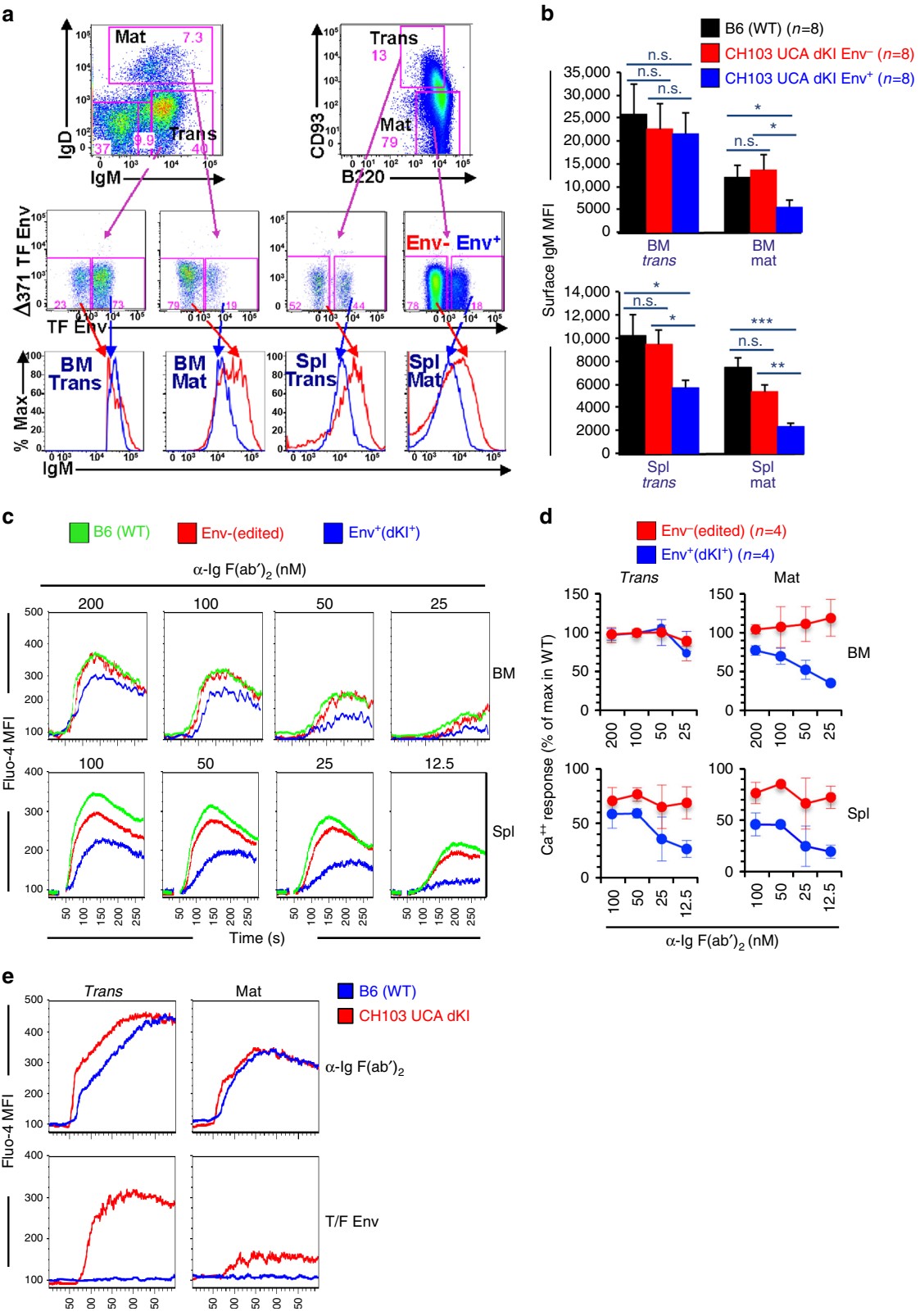

sequentially immunized animals were bona fide anergic double KI-positive clones that had been re-activated to switch and form memory. Of these double KI IgG$^+$ memory clones, somatic hypermutation (SHM) rates were low but detectable, with six out of nine having amino acid changes in their KI HC and/or LCs (Fig. 7f), but the relatively modest changes (Supplementary Fig. 10) suggest incomplete (partial) activation, and are analogous to the differential-binding population with relatively longer HCDR3s observed in macaques (Fig. 2f, g). In contrast to the high fraction of differentially binding double KI-positive clones bearing somatic mutations, none of the differentially binding clones that retained only KI HCs and just 18% of those that retained only KI LCs bore somatic mutations (Fig. 7f). This difference suggests that immunization might break anergy in rare double KI-positive clones, initiating SHM as well as driving switch and Ab secretion, while LC (or HC)-edited (non Env cross-reactive) clones (which are fully competent to signal (Fig. 6a–d)), yet are not engaged by TF Env ex vivo (Fig. 6e), are also not primed by TF Env in vivo. An alternative, non-mutually exclusive possibility is that SHM might be initiated in double KI-positive clones, to drive specificity away from self ("affinity reversion"), which would not be required in clones that have already lost expression of their KI HC or LCs.

## Discussion

We have demonstrated in this study that sequential immunization of both macaques and CD4-binding site HCDR3-binder nAb $V_H + V_\lambda$ KI mice can initiate neutralizing antibody lineages. In macaques, the sequential immunogen induced antibodies with limited neutralization breadth, likely due both in part to incomplete activation of self-reactive, lineage-specific bnAb clones as well as the unavailability of a suitable IGLV to pair with IGHV4-J for bnAb activity. In CH103 KI mice, we observed that a large proportion of antibodies used a similar $V_H$, but different $V_L$ than did the CH103 UCA, showing that receptor editing[36] was one host tolerance mechanism limiting development of CD4-binding site HCDR3-binder bnAbs. A second major tolerance mechanism in CH103 UCA KI mice was anergy in non-receptor-edited mature B cells. These data suggest that in addition to designing sequential Envs to select for the desired bnAb lineage SHMs, it will be necessary to formulate sequential vaccines with adjuvants and/or drugs to transiently circumvent immune checkpoints and overcome anergy[37].

We have previously reported that the CH103 UCA was not polyreactive using 10 autoantibody assays, whereas mature CH103 lineage antibodies acquired polyreactivity coincident with the onset of bnAb activity[6]. In a more stringent analysis of polyreactivity with a >9600 human protein chip assay, we found that indeed, the CH103 UCA was autoreactive. Thus, like the 3BNC60 UCA $V_{HL}$ KI model, expressing naive B cell precursors bearing germ line-reverted V(D)J rearrangements of the CD4-mimicking bnAb 3BNC60[17], the HCDR3-binder CD4-binding

site CH103 bnAb UCA double KI cells are also deleted in mature B cell development prior to immunization and subjected to extensive LC receptor editing. The mechanism by which editing occurred in the two models was different (lambda vs. restricted kappa usage), which could be explained either by the way the models were designed and/or by differences in stringency of editing demanded by the degree of tolerizing self-reactivity these two bnAb specificities imparted in vivo.

The CH103 UCA heterozygous double KI (het; $V_HDJ_H^{+/-}/V_\lambda J_\lambda^{+/-}$) model we used here differs in two ways from other recently reported CD4-binding site "germ line-reverted" models[15], influencing our ability to study the full bnAb lineage maturation pathway that sequential immunization will likely need to recreate. The first distinction is expression of a fully unmutated $V_HDJ_H$ rearrangement from the bona fide time-of-infection CH103 UCA, in contrast to the germ line-reverted KI models using hybrid rearrangements with germ line $V_H$ and matured bnAb HCDR3s[10, 12]. The second distinction is that the specific LC targeting strategy in our model[16, 18] accounts for more physiologically relevant tolerance mechanisms of LC receptor editing than do those of other bnAb double KI (HC + LC) KI models[38, 39]. This feature, when in combination with use of this model as a heterozygous knocked-in version at both the HC and LC loci, allowed us to evaluate physiological tolerance effects on the pre-immune repertoire. Thus, the CH103 KI mouse model represents a high bar for testing clonal competition, because a true germ line (UCA) precursor is present in a semi-polyclonal system that is most analogous to an adoptive transfer landscape.

Recent evidence for the mechanism of preferential induction of competing "off-target" CD4-binding site positive responses, at the expense of CD4-binding site positive bnAb responses has been recently reported in competition studies between precursors of CD4-binding site positive bnAb and non-nAbs using transformed B cell lines in vitro[40]. Our ex vivo Ca$^+$-signaling data in this study extend these findings and provide a direct mechanism for incomplete maturation in vivo: exclusion of anergic bnAb lineage precursors from participating in T-dependent responses at the expense of both edited and off-target clones. A model based on this mechanism predicts that bnAb clones will be at a survival disadvantage and will be outcompeted by two populations. Initially (prior to Env immunization), such anergic bnAb lineage clones will be excluded by those predominating the mature repertoire that are non Env-reactive (due to editing their LCs in response to autoantigen or environmental antigen). Such LC-edited clones, whether they completely eliminate or maintain partial self-reactivity would thus be predicted to outcompete anergic bnAb clones, since analogous clones with low-degree or "acceptable" self-reactivity, have been found to spontaneously populate the germinal centres (GC) niche in certain autoimmune mouse strains[41, 42] or autoimmune conditions[43]. Then later, in an immunization setting involving Env, a highly complex, multi-epitope immunogen (that even in trimeric form has a high

**Fig. 6** Signaling responsiveness of transitional and mature B cell subsets from CH103 germ line knock-in mice. **a** Representative flow histograms showing selectively reduced IgM mean fluorescence intensity (MFI), in differential-binding transmitted-founder (TF) Envelope (Env$^+$) mature B cell fractions in CH103 unmutated common ancestor (UCA) double knock-in (dKI) mice, relative to Env$^-$ (light-chain (LC)-edited) fractions. Data show gating for Env-specific bone marrow (BM) and splenic Env$^+$ and Env$^-$ trans or mat B cell fractions, and is gated on live, total (B220$^+$) B cells or live, total splenocytes, respectively. **b** Graphical summary (means + SEM) of data shown in (**a**) performed on eight mice per strain. **c** Ex vivo proximal signaling responses to BCR cross-linking in naive B cells from CH103 UCA dKI and control B6 mice, based on Ca$^{++}$ levels (Fluo-4 MFI) before/after anti-IgM stimulation. Shown are representative data (from a single CH013 UCA dKI mouse) of Ca$^{++}$ responses, to varying concentrations of anti-BCR, in mature B cells, either recirculating in BM (upper row) or in spleen follicles (lower row). Also shown for comparison is proximal signaling of mature B cells from control WT B6 mice. **d** Graphic representation of Ca$^{++}$-signaling responses (means $\pm$ SD) in both transitional and mature (splenic or BM B cells). Data are cumulative from four mice and two independently performed experiments. **e** CH103 UCA dKI transitional and mature B cell subset responses to priming immunogen TF Env. Data shown are from experiments performed in BM B cell fractions, using saturating amounts (100 nM) of tetramerized TF Env gp120 or anti-BCR (as a positive control for 100% of B cells signaling). *$P < 0.05$; **$P < 0.005$, ***$P < 0.001$; two-tailed Student's $t$-test. n.s. not significant

capacity for triggering off-target clones), anergic bnAb B cell clones may be further excluded from GCs by non-anergic, off-target (that is, non-differential Env$^+$) clones induced by Env vaccination. We predict that such clones will begin arising at high frequencies in memory IgG populations after multiple immunizations and after numerous rounds of preferential selection by immunization. Their presence will make it difficult for bnAb lineage clones to compete for the GC niche and form long-lived memory, thus requiring much stronger local CD4 helper and/or BCR signals to overcome this hurdle.

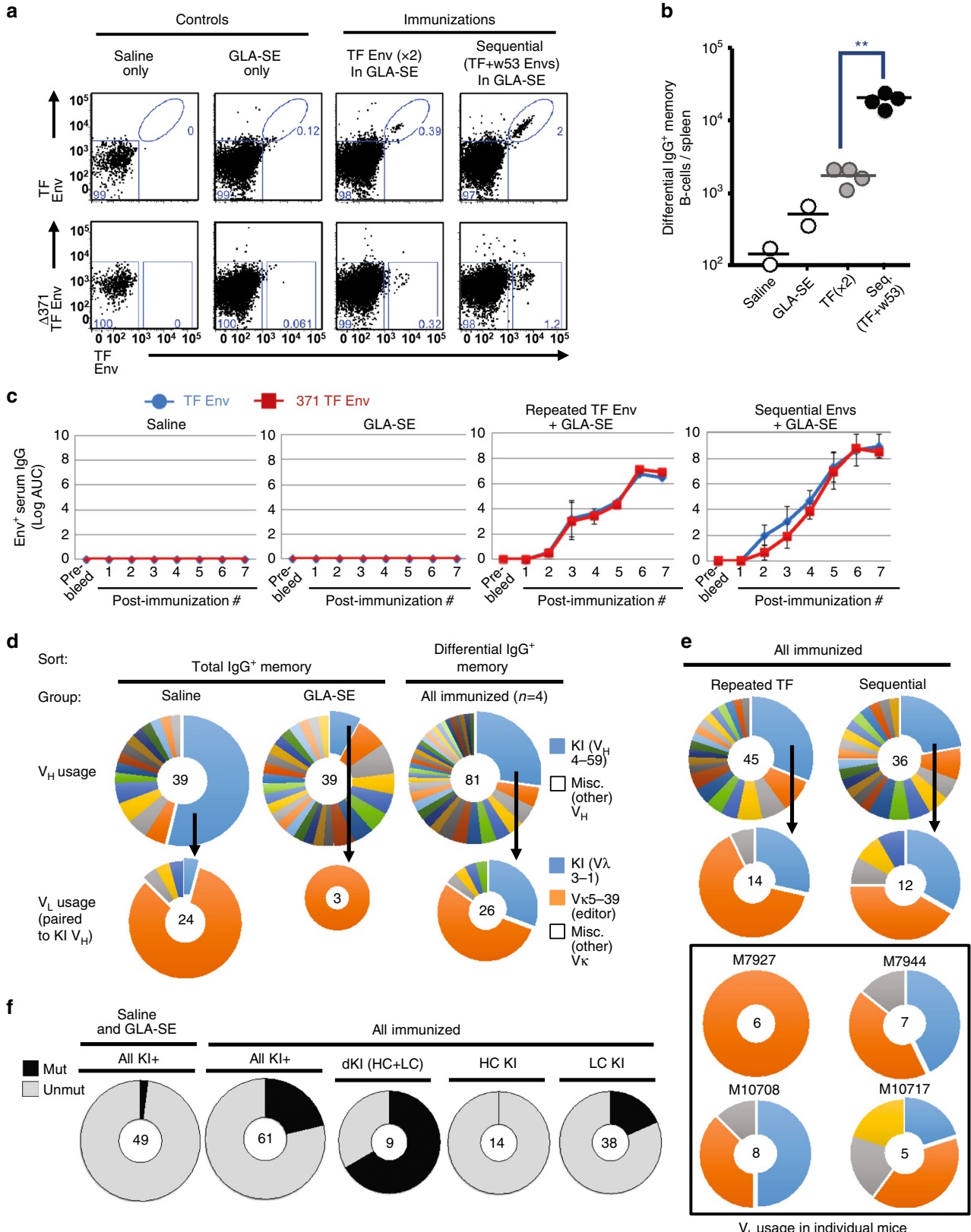

A key question is why TF followed by week 53 Env led to more differential Env binding (lineage-positive/unedited) B cells, that is, enriched for double KI clones, expressing the original UCA $V_HDJ_H/V_JJ_L$ pair. Although we do not have direct evidence for the mechanism behind this, we do not think week 53 Env directly rescued such differential Env-binding B cells, since we do not think week 53 Env has high enough affinity for the UCA to activate anergic B cells expressing the UCA. However, week 53 Env may indirectly help "break" anergy in double KI clones by: (i) diverting the non-lineage positive B cell response, thus reducing competition with anergic clones; this would specifically occur via week 53 Env potentially cross-reacting more strongly than TF with either pre-immune KI light-chain-edited (TF Env−) clones and/or vaccine-induced KI heavy-chain-edited (TF Env+ but non-differential) clones that start appearing after TF priming; or (ii) activating/expanding the rare double KI-positive clones initially activated by TF to switch and expand, because, as shown in this study, such clones exhibit more SHM (albeit limited amounts) than heavy- or light-chain-only KI clones, which could produce a relatively higher fraction of clones with affinity for week 53 Env.

In conclusion, future studies should focus on overcoming the factors that limit induction of CH103 bnAbs, by testing new CH505 Env vaccine regimens in CH103 KI mice and in macaques. These regimens should increase the precursor pool, by selection for non-LC-edited clones, and during immunization, they should enhance the strength of in vivo Tfh and/or BCR signals to the lineage-specific anergic clones, in order to provide a survival advantage to bnAb B cell lineages over "off-target" B cell lineages. It will also be critical to test sequential CH505 Env Clade C gp120 immunogens in humans, since macaques do not have orthologs of the human genes that encoded the CH103 bnAbs induced by CH505 Envs.

## Methods

**Study design.** We previously identified a transmitted-founder (TF) clade C (CH505 TF) HIV-1 envelope (Env) that induced CD4-binding site bnAbs CH103 and CH235 in an HIV-1-infected African individual (CH505)[6]. Ab-virus co-evolution studies in the CH505 HIV-1-infected individual revealed that bnAbs arose after extensive Env diversification. Using this roadmap of Ab-viral co-evolution, we selected a 4-valent CH505 Env regimen and produced them as gp120 candidate Env immunogens for vaccination in adult non-human primates (NHPs) and CH103 UCA KI mice. We hypothesized that the CH505 Envs would induce CD4-binding site, CH103−, and/or CH235-like nAbs. Thus, to test the immunogenicity of the CH505 Env immunogens, we immunized HIV-1 uninfected rhesus macaques. Furthermore, to test the ex vivo and in vivo immunogenicity and specificity of CH505 Envs used in macaques, and to define the host mechanisms impacting the development of HCDR3-binder, CD4-binding site antibodies both prior to, and in response to immunization, we generated CH103 UCA heavy-chain and/or light-chain-expressing KI mice and evaluated the immune response to CH505 Env regimens used in macaques.

**Ethics.** Indian-origin rhesus monkeys used in the immunization studies (NHP79 and NHP88) were housed and maintained in an Association for Assessment and Accreditation of Laboratory Animal Care-accredited institution in accordance with the principles of the National Institute of Health. All studies were carried out in strict accordance with the recommendations in the Guide for the Care and Use of Laboratory Animals of the National Institutes of Health in BIOQUAL (Rockville, MD). BIOQUAL is fully accredited by AAALAC and through OLAW, Assurance Number A-3086. The animal protocol used in this study was approved by the BIOQUAL IACUC. All physical procedures associated with this work were done under anesthesia to minimize pain and distress in accordance with the recommendations of the Weatherall report, "The use of non-human primates in research." Teklad 5038 Primate Diet was provided once daily by animal size and weight. The diet was supplemented with fresh fruit and vegetables. Fresh water was given ad libitum. Mamu-A*01-negative Indian-origin rhesus monkeys were housed at Bioqual, Inc., Rockville, MD. All monkeys were maintained in accordance with the Guide for the Care and Use of Laboratory Animals. All animal protocols were approved by the Institutional Care and Use Committee.

For all mouse work in this study, including characterization of the various naive CH103UCA KI models in relation to wild-type (WT) B6 (background, gender, and age-matched) controls, as well as immunization studies in CH103 double KI mice, all aspects of the procurement, conditioning/quarantine, housing, colony management, veterinary care, and carcass disposal programs comply with the guidelines set forth by the NIH guide for the care and use of laboratory Animals, the Animal Welfare Act, and all applicable federal, state, and institutional laws. All mice for these studies were housed in the Medical Sciences Research Building II Vivarium at Duke University Medical Center (DUMC) in a pathogen-free environment with 12-h light/dark cycles at 20–25 °C, in accordance with Duke University Institutional Animal Care and Use Committee-approved animal protocols. In addition, all mice in the MSRBII vivarium are also frequently tested for murine pathogens by culture, histopathology, and serology.

**Recombinant Env gp120 expression.** One mg of plasmid DNA per 1 liter of cells was diluted in DMEM and mixed with PEI. PEI:DNA mixtures were added to 293F cells (ThermoFisher, catalog #R79007) for 4 h. 293F cells were subsequently washed and diluted to 1.25 million cells per mL in Freestyle293 media (ThermoFisher). The cells were cultured for 5 days, and on the fifth day, the cell culture media was cleared of cells by centrifugation and filtration with a 0.8 μm cellulose membrane (Nalgene). The cell culture media was concentrated with a vivaflow 50 with a 10 kD molecular weight cutoff. The concentrated cell culture supernatant was rotated with lectin beads (Vistar Labs) in MES pH 7.0 buffer overnight at 4 °C. The beads were pelleted by centrifugation the next day and resuspended in MES pH 7.0 wash buffer. The lectin beads were washed twice and the glycosylated HIV-1 gp120 was eluted with 0.5 M methyl-α-pyranoside. The protein was buffer-exchanged into phosphate-buffered saline and stored at −80 °C. Monomeric recombinant gp120 was purified by size exclusion chromatography (SEC) in phosphate-buffered saline using a Superdex200 10/300 gel filtration column (GE Healthcare) and stored at −80 °C. The glycans on recombinant gp120 Envs have been shown to be more complex residues compared to glycans on virus associated gp160 Envs[47].

**HIV-1 CH505 Env immunizations.** Previous studies have shown the adjuvants, synthetic monophosphoryl lipid A in stable emulsion (GLA-SE)[48], and liposome-based formulation with MPL and QS21 (AS01E)[49], used in this study to be effective for enhancing humoral immune responses to a number of infectious agent proteins. CH505 TF gp120 Envs were selected as Env immunogens based on affinity for intermediate (IA) and mature antibodies in the CH103 lineage[20]. In addition, we used surface plasmon resonance (SPR) analysis to determine the affinity of CH103 lineage antibodies to the 4 CH505 gp120 Envs selected for our immunization regimens. Each CH103 lineage mAb binding to CH505 Envs (TF, week 53, week 78, week 100) was measured by injecting Env proteins at varying concentrations (0.5–25 μg/mL) over each mAb captured on anti-rhesus IgFc immobilized Ab (Millipore) on a CM5 sensor surface[50] with the T200 SPR platform. We found that CH103UCA bound only autologous CH505 TF gp120 (521 nM), while

**Fig. 7** Early (TF + week 53) phase of Env "4-valent" sequential immunization rescues unedited (anergic) mature B cells in CH103 germ line knock-in mice. **a** Representative FACS dot plot histograms of IgG-switched splenic memory B cells in control or experimental groups denoted. Shown is transmitted-founder (TF) Env reactivity of splenic IgG+ memory B cells, either gated for wild-type (WT) TF Env (non-differential; upper row) or differential, that is, lineage-specific (Δ371 TF− vs. WT TF+; lower row) gating strategies. Data are shown after second immunizations in CH103 unmutated common ancestor (UCA) heterozygous (het) double knock-in (dKI) ($V_HDJ_H^{+/−} \times V_\lambda J_\lambda^{+/−}$) mice. Gating was performed on live, class-switched memory (B220+ CD19+ CD93− CD38+ IgG+ IgM−) B cells. **b** Graphical representation of differential-binding IgG+ memory B cells per mouse spleen. Circles represent individual mice, and means are denoted by black bars. **P < 0.005, two-tailed Student's t test. **c** Area under the curve (AUC) (mean + SEM) ELISA binding of serum IgG+ from TF Env-immunized mice, to WT or mutant Δ371 TF Env. Sera was collected 10 days after each immunization. Note the low-titer, differential serum IgGs early, but higher-titer, non-differential IgG Ab responses later in the sequential regimen. Also note that the last two boosts were carried out using the week 100 Env. **d**, **e** Distribution of CH103 prec. human $V_H$4-59 and $V_\lambda$3-1 segments vs. other, endogenous murine $V_H$ and $V_\kappa$ families utilized, in differential-sorted splenic IgG+ CD38+ memory B cells from immunized mice. Shown as comparisons are single IgG+ cells sorted from control-immunized mice. Middle circles; number of single-cell pairs sequenced. Relative sizes of LC pie charts indicate the % of LCs pairing to the KI HCs, as a proportion of total single-cell clone numbers amplified/sequenced. **f** Amino acid mutation status of memory IgG+ clones, pooled from all immunized CH103 UCA het dKI mice (n = 4), shown as all pairs retaining a residual heavy-chain (HC) and/or KI allele, or broken down into three groups: either as dKI clones, or as HC-KI-only or light-chain (LC)-KI-only-bearing clones

CH103 intermediate and mature Abs bound with higher binding affinity to CH505 TF and natural variants: CH505 TF (CH103_IA3, 23 nM; CH103_IA2, 12.4 nM; CH106, 9.4 nM); CH505 week 53 (CH103_IA3, 135.6 nM; CH103_IA2, 50.2 nM; CH106, 30.2 nM); CH505 week 78 (CH103_IA3, 96.3 nM; CH103_IA2, 30.9 nM; CH106, 12.7 nM); and CH505 week 100 (CH103_IA3, 91.2 nM; CH103_IA2, 15 nM; CH106, 12.3 nM).

We immunized a total of 44 macaques in eight groups with CH505 transmitted-founder (TF) or natural variants of the TF (week 53, week 78, week 100) envelope (Env) alone, or sequential combinations of TF and TF-variants in AS01E (NHP88 study) or GLA-SE (NHP79 study) adjuvants. Of 44 macaques aged 3–4 years, 4 (5628, 5644, 5714, and 5723) were female and the remaining 40 were males. Immunizations were administered intramuscular in the leg of each animal with 100 µg of total CH505 gp120 proteins for six vaccinations. For macaques immunized with CH505 gp120 Envs in AS01E (NHP88), animals were immunized on a 6-week interval for the first 4 immunizations, followed by a 7.5-month interval until boost #5 and a 3-month interval from boost #5 to #6. Vaccine group 1 animals received sequential CH505 gp120 Env regimen (+AS01E) as follows: IMM #1—TF, IMM #2—week 53, IMM #3—week 78, IMM #4—week 100, and IMM #5 and #6—week 100. In contrast, vaccine group 2 animals received CH505 TF gp120 Env alone (+AS01E) at all immunizations (IMM #1–6—TF). For macaques immunized with CH505 gp120 Envs in GLA-SE (NHP79), animals were immunized on a 6-week interval for the first 5 immunizations, followed by a 8.2-month interval until boost #6. Vaccine group 3–6 animals received only one CH505 gp120 Env (+GLA-SE) for all immunizations (IMM #1–6); group 3 (TF), group 4 (week 53), group 5 (week 78), and group 6 (week 100). In contrast, vaccine groups 7–8 animals received sequential combinations of CH505 gp120 Envs (+GLA-SE): group 7_IMM #1—TF, IMM #2—week 53, IMM #3—week 78, IMM #4—week 100, IMM #5—TF + week 53 + week 78 + week 100, and IMM #6—week 100; group 8_IMM #1—TF, IMM #2—TF + week 53, IMM #3—TF + week 53 + week 78, IMM #4—TF + week 53 + week 78 + week 100, IMM #5—TF + week 53 + week 78 + week 100, and IMM #6—week 100.

For all immunizations with CH103 KI models, a minimum of 4 mice per immunization group (genders distributed equally) were used, all were 8–12 weeks old at the start of the immunization study. Mice with all Duke University Institutional Animal Care and Use Committee-approved animal protocols. Mice were immunized up to 8 times with saline (control groups), 5 µg of the TLR4 agonist-based, Infectious Disease Research Institute proprietary adjuvant system GLA-SE (adjuvant control groups), or with 5-µg GLA-SE and 25 µg of various CH505 Env gp120 proteins (UCA-targeting TF alone, or with additional CH505 lineage intermediate Ab-binding Env variants week 53, week 78, week 100), via intraperitoneal injections (200 µl) administered every 21–28 days. Blood samples were collected for isolation of sera 10 days after each immunization.

**Macaque memory B cell single-cell sorting and phenotyping.** We performed single-cell isolation of macaque memory B cells decorated with both AlexaFluor® 647 (AF647) and Brilliant Violet 421 (BV421)-tagged HIV-1 CH505 TF gp120 using a BD FACSAria™ or a BD FACSAria™ II (BD Biosciences, San Jose, CA), and the flow cytometry data were analyzed using FlowJo (Treestar, Ashland, OR)[51]. Sorting strategies included dual cell staining of AF647 and BV421-tagged CH505 TF gp120 proteins (CH505 double positives), and single-cell staining of BV421-tagged CH505 TF gp120, but not AF647-tagged CH505 TF gp120 Δ371-mutant protein (CH505 single positives). Single-cell sorted Env-reactive memory B cells were CD3 (BD #552852; 2.5 µl per test)/CD14 (BioLegend #301832; 5 µl per test)/CD16 (BD #557744; 5 µl per test) and/or surface IgD (Southern Biotech #2030-09; 1 µl per test) negative, but CD20 (BD #347673; 5 µl per test) positive. Antibodies were used to stain $1 \times 10^6$ peripheral blood memory cell (PBMCs) per test for single-cell sorting. In addition, BV421-tagged CH505 TF gp120 and AF647-tagged CH505 TF gp120 Δ371-mutant protein were used to decorate $1 \times 10^6$ macaque PBMCs from which IgD and CD27-All memory B cells were phenotyped using BD FACSAria™ (BD Biosciences), and the flow cytometry data were analyzed using FlowJo (Treestar). The following Abs were used for memory B cell phenotyping: CD20 FITC (BD Biosciences, catalog #347673), 10 µl; IgD PE (Southern Biotech, catalog #2030-09), 2 µl; CD8 PE-TxRed (Invitrogen, catalog #MHCD0817), 3 µl; CD16 PE-Cy7 (BD Biosciences, catalog #557744), 1 µl; IgM PE-Cy5 (BD Biosciences), 20 µl; CD27 APC-Cy7 (BioLegend), 1 µl; CD14 BV570 (BioLegend, catalog #301832), 5 µl; and CD3 PerCP-Cy5.5 (BD Biosciences, catalog #552852), 5 µl. Macaque B-cell phenotyping gates for populations studied were determined based on the single Env-stained controls (cells stained with BV421-tagged CH505 TF gp120 alone or AF647-tagged CH505 TF gp120 Δ371), but background staining detected at week 0 in some samples. The positivity cutoff for macaque memory B-cell phenotyping was defined as mean + 3 SD at immunization number: CH505 differentials (sequential and TF Env vaccines)—0.03, CH505 non-differentials—0.26 (sequential Env vaccines), 0.08 (TF Env vaccine).

**Single-cell sorting of memory B cells from CH103 UCA KI models.** For single-cell sorting of activated (class-switched, memory) B cells from CH103 UCA KI models, single suspensions of total splenocytes and/or draining lymph nodes were stained using 0.5 µg/mL of anti-B220 BV650 (catalog #563893), anti-CD19 APCR700 (catalog #565473), anti-CD93 PE-CF594 (catalog #563805), anti-CD38 BV786 (catalog #740887), anti-IgD BV510 (catalog #563110), and anti-IgG1 FITC

(catalog #553443), G2a/2b FITC (catalog #553399), G3 FITC (catalog #553403); all Abs were obtained from BD Biosciences. IgG+ memory B cells were visualized by first gating on lymphocyte singlets, followed by exclusion of LIVE/DEAD® near infrared-stained cells (to discriminate live from dead cells), subsequent gating for CD19+B220+ (total B cells), and finally, gating for CD38+IgG+IgD− (class-switched, memory B cells). CH103 lineage specificity of the IgG+ memory B cell subset was detected by using the combination of tetramerized, AlexaFluor 647, and Brilliant Violet 421-tagged HIV-1 Env gp120 wild-type or mutant CH505 transmitted/founder virus-derived proteins (TF-BV421 and Δ371 TF-AF647, respectively), with differential binding to the former, but not the latter, indicating CH103 bnAb lineage-specific binding to the Env CD4-binding site, as previously described[6]. Differential-binding (TF+, Δ371 TF−), switched memory (near IR−, CD19+B220 +CD38+IgG+IgD−) B cells induced in Env-immunized mice, and for comparison, non-HIV Env binding (WT TF−, Δ371 TF−) switched B cells from naive, control (saline or "adjuvant alone")-immunized mice, were sorted using a FACSAria II (BD Biosciences) into BioExpress 96-well plates (T-3085-1) containing 20 µl of SuperScript® III reverse transcriptase buffer (LifeTech) as previously described[52]. Sorted plates were frozen in a dry ice ethanol bath and stored at −80 °C until further processed.

**Mouse B cell phenotypic analysis by flow cytometry.** Flow cytometric analysis of B cell development and responses was performed as described[14, 16, 18, 19]. Briefly, single-cell suspensions from spleen, draining lymph nodes (dLNs), and BM were isolated from 8–12-week-old naive CH103UCA $V_HDJ_H^{+/+}$ KI, $V_HDJ_H^{+/−}$ KI, double KI, het double KI mice, and WT naive C57BL6 mice (used as controls), or immunized CH103UCA het double KI mice. A total of $10^7$ cells were first stained with LIVE/DEAD staining buffer (LifeTech), spun down, and then stained in FACS buffer containing $1 \times$ PBS (pH 7.2), 3% FBS (Hyclone), 0.01% sodium azide, and premixed combinations of fluorochrome-labeled mAbs at titration-determined optimal concentrations. Total B cells were gated as singlet, live, CD19+B220+ lymphocytes. Primary labeled mAbs (all from BD Biosciences) used were as follows: 0.5 µg/mL of anti-B220 BV650 (catalog #563893), anti-CD19 APCR700 (catalog #565473), anti-IgD BV510 (catalog #563110), anti-IgG1 FITC (catalog #553443), anti-IgG2a/2b FITC (catalog #553399), anti-IgG3 FITC(catalog #553403), anti-IgM PE-Cy7 (catalog #552867), anti-CD21 BV421 (catalog #562756), anti-CD23 FITC (catalog #553138), and anti-CD93 PECF594 (catalog #563805); and 0.2 µg/mL of anti-mouse T and B cell activation antigen PE (catalog #561530) and anti-Fas PE-Cy7 (CD95, catalog #557653). Flow cytometric analysis of subset B cell reactivities for CH103 lineage-specific CD4-binding site specificity was performed similarly using single-cell splenocyte, dLN, or BM suspensions from naive and immunized mice that were stained with fluorochrome-labeled wild-type and mutant (Δ371) CH505 TF Env gp120 tetramers, also as previously described[6].

**Isolation of macaque antibody genes and evaluation of immunogenetics.** Heavy-(IGHV) and light (IGKV, IGLV)-chain genes were isolated via single-cell PCR[52, 53]. The gene sequences were then computationally analyzed and immunogenetics (gene family and segment IDs, mutation frequency, CDR3 length) determined using Cloanalyst program[54–56]. Antibody sequences were studied using the 2015 Cloanalyst rhesus gene library. Antibody clonal lineages were inferred as sequences that had the same IGHV VDJ rearrangement and CDR3 length, and paired with the same light chain (Ig VJ segments). The automated inference was followed up by visual inspection of the DNA sequence alignments for confirmation. The heavy- and light-chain gene sequences for the inferred UCA and IAs were produced commercially and used to generate purified recombinant mAbs.

**Isolation of mouse antibody genes.** Heavy-chain (HC) $V_HDJ_H$ and LC $V_LJ_L$ rearrangement pairs from single-sorted memory B cells were recovered via PCR based on previous methods[16, 18, 57–59]. Briefly, complementary DNAs from individual 96-well plates were generated by reverse transcription synthesis using SuperScript® III Reverse Transcriptase and random primers as per manufacturer's instructions (LifeTech), followed by two rounds of PCR amplification using Ig reverse and forward primer sets. For isolation of KI HC rearrangements, forward primers (outer and nested) were used that were specific for either the common J558 H10 leader peptide in the KI HC expression cassette[59] or human $V_H4$-59. Parallel PCR amplifications using degenerate leader $V_H$ family-specific primer mixtures[58] were also used to detect mouse endogenous HC rearrangements. For all HC rearrangements isolated (KI or endogenous), reverse primer mixtures of Cγ1, Cγ2b, Cγ2c, Cγ3, and Cα-specific primers[58] were used to specifically isolate only activated (i.e., class-switched) B cells. Likewise, for isolation of LC rearrangements, forward primers specific to either the common VκOx1 leader peptide in the KI LC expression cassette[16, 18] or human $V_λ$3-1 gene-specific were used, whereas forward degenerate leader Vκ[57] and $V_λ$ primer mixtures, in combination with reverse Cκ and $C_λ$-specific primers, respectively, were used to detect endogenous LC kappa[57, 60] or LC lambda[57] rearrangements, the former which could also occur as 2° rearrangements on the knocked-in allele (to replace the original CH103UCA $V_λJ_λ$). Cloned PCR products were then purified and directly sequenced in both orientations (GeneWiz) using published primers[16, 18, 59], and V, D, and J segment

usage was determined by querying amplified sequences to both the original CH103UCA rearrangements and relative to C57BL/6 Ig germ line sequences in the IMGT database using IMGT/HighV-QUEST search software. CH103UCA KI $V_HDJ_H$ and $V_\lambda J_\lambda$ KI rearrangements sequenced in both directions were analyzed for SHMs using DNASTAR MegAlign Pro multiple sequence alignment software.

**Expression of macaque antibody genes as IgG1 recombinant mAbs.** Plasmids encoding the *IGHV*, *IGKV*, and *IGLV* genes were generated and used for recombinant mAb production in human embryonic kidney epithelial (HEK) cell lines 293T (ATCC, Manassas, VA; catalog #CRL3216) in small-scale transfection[52], and in suspension Expi 293F cells (Invitrogen; catalog #A14527) for expression of larger quantities of purified mAbs[6]. Purified recombinant mAbs were dialyzed against PBS, analyzed, and stored at 4 °C.

**Antibody binding.** Plasma antibody reactivity with CH505 Envs was determined via standard enzyme-linked immunosorbent assays (ELISA) and binding titers reported as log area under the curve (AUC)[61]. Recombinant mAbs and plasma antibodies were screened for reactivity with HIV-1 Envs and corresponding mutant proteins with a disruption of the CD4-binding site in ELISA[61]; these proteins included CH505 TF gp120, CH505 TF gp120 Δ371I, YU2 gp120, YU2 D368R gp120, RSC3, RSC3 Δ371, and RSC3 P363N-Δ371I proteins[6, 62]. Abs that bound CH505 TF gp120, but not CH505 TF gp120 Δ371, are candidate CD4-binding site antibodies referred to as CH505 differentials; CH505 non-differentials bound equally well to both proteins.

**Antibody-competitive inhibition.** Plasma and purified recombinant mAbs were evaluated for blocking well-characterized antibodies as previously described[50]. For CD4 (binding site) blocking assays, we used sCD4 (Progenics Pharm Inc.) or CH106 bnAb[6]. Competitive inhibition was measured as the ratio of binding in the presence and absence of inhibitory molecules, CH106 or sCD4.

**Neutralization assays.** Plasma and purified mAbs were screened for neutralization using the well-established TZM-bl assay as described[63]. Plasma post sixth/final immunization neutralized autologous tier 1 (CH505.w4.3), and heterologous tier 1 (MW965, SF162, 6644, DJ263) and two (57128) viruses. Heterologous tier 2 viruses tested for which we did not observe plasma neutralization are as follows: CON-S, 45_01dG, JRFL, YU2, ZM1735P, Q842, Q168, and BG1168. DH522 lineage mAbs were tested for neutralization breadth in a panel of 199 geographically diverse viruses as described[64]; DH522 mAb neutralized viruses 6095.V1.C10, DJ263.8, 6535.3, ADA.DG, BaL.26, BX08.16, HXB2.DG, MN.3, SF162.LS, SS1196.01, CNE40, 57128.vrc15, 6644.V2.C33, and MW965.26.

**Serum ELISA assays for CH103 UCA double KI model.** Titers of class-switched serum antibodies reactive for the wild-type CH505 TF Env protein, or a mutant version with a disruption of the CD4-binding site (TF Δ371), was determined by ELISA, based on described methods[6]. Briefly, mouse sera, collected 10 days after each immunization, were applied to ELISA plates coated with CH505 wild-type or Δ371-mutant TF Env gp120 proteins, in 12-point serial dilutions starting at 1:30, in order to determine AUC concentrations. -Alkaline phosphatase-conjugated goat anti-mouse IgG antibodies (Southern Biotech) was applied and detected using Tetramethylbenzidine (TMB) substrate/acidic stop solution and read on a standard laboratory spectrophotometer at 450 nm.

**Calcium flux analysis in CH103 UCA double KI model.** For experiments evaluating total splenic B cells, WT BL/6 and CH103 UCA double KI splenocytes were collected, and total B cells were enriched using a mouse Pan-B cell isolation kit (Stemcell) according to manufacturer's instructions. Enriched Pan-B cells were stained by LIVE/DEAD® Fixable Yellow Dead Cell Stain Kit (ThermoFisher Scientific) for 30 min. For experiments evaluating BM and splenic B cell subsets, single-cell suspensions were directly stained with various combinations of cell surface markers for BM fractionation into pre-B, immature-B, transitional B, and mature B subsets included 0.5 μg/mL of anti-B220 BV786 (catalog #563894), anti-CD19 APCR700 (catalog #565473), anti-CD43 BV605 (catalog #563205), anti-IgM PE-Cy7 (catalog #552867), and anti-IgD BV605 (catalog #563003). Likewise, cell surface marker combinations used for spleen subfractionation into transitional-B and mature-B subsets included 0.5 μg/mL of anti-B220-BV786 (catalog #563894), anti-CD19 APCR700 (catalog #565473), and anti-CD93 BV650 (catalog #563807). All subsets were also further subdivided for CD4-binding site⁺ CH103 lineage reactivity by including AF647 and BV421-fluoresceinated CH505 TF Env gp120 tetramers in the pre-stain mix. For both sets of experiments, pre-stained B cells were loaded with Fluo-4 via thorough washes in HBSS, followed by mixing with equal volumes of 2× Fluo-4 Direct™ calcium reagent loading solution (Fluo-4 Direct™ Calcium Assay Kits, ThermoFisher Scientific). After sequential 30 min incubations at 37 °C and room temperature, cells were washed and incubated with LIVE/DEAD® staining buffer for 30 min. and resuspended in calcium-containing HBSS and incubated at room temperature for 5 min, before being activated by 25 μg/mL anti-IgM F(ab')₂

(Southern Biotech). Fluo-4 MFI data for transitional (B220⁺CD93⁺) B cells was acquired on a BD LSR II flow cytometer and analyzed by FlowJo software.

**Crystallography.** Fab fragments of DH522UCA, DH522IA, DH522.1, and DH522.2 were produced recombinantly as previously described[65]. Briefly, Fab chains were generated by cloning into pcDNA3.1 (+)/hygro, the synthesized variable region of a heavy-chain gene upstream of the CH1 IgG1 domain[66]. The Fab fragment of the heavy chain was transfected with the corresponding full-length light chain into 293i cells using Expifectamine (Invitrogen) and purified with lambda select or kappa select resin (GE Healthcare) as previously described[65]. Fabs were further purified via SEC using a HiLoad 26/60 Superdex 200 pg 26/60 column at 3.2 mL/min with a buffer of 10 mM Hepes pH 7.2, 50 mM NaCl, 0.02% NaN₃. Peak protein-containing fractions were concentrated, buffer-exchanged to ddH₂O, and brought to 15.0 mg/mL. A gp120 core with chimeric B.YU2 sequence[67] was produced by transfecting HEK 293 s GnTI⁻/⁻ cells (ATCC, Manassas, VA; catalog #CRL3022) using PEI as the transfection reagent. Expressed glycosylated gp120 core was purified from the cell culture supernatant with lectin beads in pH 7.0 MES buffer (Sigma; St Louis, MO), washed and eluted with Methyl-Manno-Pyranoside (0.5 M) (Sigma) in pH 7.0 MES buffer. The gp120 core was buffer exchanged into phosphate-buffered saline and stored at 4 °C until being further purified via SEC as above. The purified, glycosylated protein was treated with Endo H, at 1000 U per μg gp120 in stock reaction buffer (NEB) at a final gp120 concentration of 9.9 mg/mL gp120 and incubated overnight at room temperature with agitation. The deglycosylated protein was then run through SEC as described above. The complex of DH522.2 Fab with deglycosylated chimeric B.YU2 gp120 core was formed by binding gp120 with Fab in a 1:1.2 molar ratio, respectively, and then run through SEC as described above to eliminate excess Fab from the 1:1 complex. Peak protein-containing fractions were concentrated to 15.1 mg/mL in the SEC buffer. All protein samples were tested against commercially available screens (Qiagen, Molecular Dimensions) in SBS format sitting drop plates via automation (Douglas Instruments Ltd) with 60 μl reagent reservoirs and drops composed of 0.2 μl proteins with 0.2 μl reservoirs. DH522UCA was crystallized over a reservoir of 1 M LiCl, 0.1 M citric acid pH 4, 20% PEG 6000. DH522IA was crystallized over a reservoir of 3.6 M sodium formate and 10% glycerol. DH522.1 Fab was crystallized over a reservoir of 0.095 M sodium citrate pH 5.6, 19% isopropanol, 19% PEG 4000, and 5% glycerol. DH522.2 Fab was crystallized over a reservoir of 0.075 M Tris-HCl pH 8.5, 1.5 M ammonium sulfate, 25% glycerol. Crystals of DH522.2-gp120 complex were observed over a condition of 0.1 M MES pH 6.5, 12% PEG 20,000. All crystals were briefly soaked in reservoir supplemented with 10% ethylene glycol except unliganded DH522IA Fab, which already had glycerol in the drop. Crystals were then flash-frozen in liquid nitrogen. Diffraction data were collected at SER-CAT or in-house with an incident beam of 1 Å in wavelength. Data were reduced in HKL-2000[68]. The DH522.1 Fab structure was phased by molecular replacement in PHENIX[69] using source models chosen by high-sequence homology: the heavy chain of CH103 UCA Fab[70] and light chain of A32 Fab[71] composited together via superposition into Fv and Fc domains. Molecular replacement was performed for the other antibodies in the DH522 lineage using the unliganded DH522.1 Fab structure as the source model. For the DH522.2-gp120 complex structure, the search model for the gp120 component was drawn from the complex structure of antibody 2.2c with YU2 gp120 envelope and CD4[72]. For all structures, rebuilding and real-space refinements were done in Coot[73] with reciprocal space refinements in PHENIX[74] and validations in MolProbity[75].

**Electron microscopy.** To generate the autologous HIV-1 CH505 SOSIP.664 expression construct, we followed established SOSIP design parameters[76]. Briefly, the CH505 SOSIP.664 trimer was engineered with a disulfide linkage between gp120 and gp41 by introducing A501C and T605C mutations (HxB2 numbering system) that covalently links the two subunits of the heterodimer. The I559P mutation was included in the heptad repeat region 1 (HR1) of gp41 for trimer stabilization, and a deletion of part of the hydrophobic MPER, in this case residues 664–681 of the Env ectodomain. The furin cleavage site between gp120 and gp41 (508REKR511) was altered to 506RRRRRR511 to enhance cleavage. The resulting, codon-optimized *CH505 SOSIP.664 env* gene was obtained from GenScript (Piscataway, NJ) and cloned into pVRC-8400 using Nhe1 and NotI restriction sites and the tissue plasminogen activator signal sequence. DH522.2 Fab was expressed using transient transfection of HEK 293F suspension cells using linear polyethylenimine (PEI) following the manufacturer's suggested protocol. After 5 days of expression, supernatants were clarified by centrifugation. The clarified supernatant was diluted twofold using 1× PBS buffer and purified using CaptureSelect LC-lambda (Hu) affinity matrix (Thermo Fisher Scientific), according to manufacturer's protocols. Fractions containing the protein of interest were pooled, concentrated, and further purified by gel filtration chromatography in buffer A using a superdex 200 analytical column (GE Healthcare) in a buffer of 2.5 mM Tris, pH 7.5, 350 mM NaCl, and 0.02% sodium azide. CH505 SOSIP.664 was transfected together with a plasmid encoding the cellular protease furin at a 4:1 Env:furin ratio in 293F cells. The cells were allowed to express the soluble trimer for 5–7 days. Culture supernatants were collected and cells were removed by centrifugation at 3800×g for 20 min, and filtered with a 0.2 μm pore size filter. The soluble SOSIP was purified by flowing the supernatant over a lectin (Galanthus nivalis) affinity chromatography column overnight at 4 °C. The lectin column was washed with 1× PBS, followed

with 1× PBS supplemented with 0.5 M NaCl and proteins were eluted with 1 M methyl-α-D-mannopyranoside dissolved in 1× PBS. The eluate was concentrated and loaded onto a Superdex 200 10/300 GL column (GE Life Sciences) pre-equilibrated in a buffer of 5 mM HEPES, pH 7.5, 150 mM NaCl, and 0.02% sodium azide for analysis by electron microscopy (EM). Purified CH505 SOSIP.664 trimer was incubated with a five molar excess of DH522.2 Fab at 4 °C for 1 h. A 3 μL aliquot containing ~0.01 mg/mL of the Fab—CH505 SOSIP.664 complex was applied for 30 s onto a carbon-coated 400 Cu mesh grid that had been glow discharged at 20 mA for 30 s, followed by negative staining with 2% uranyl formate for 20 s. Samples were imaged using a FEI Tecnai T12 microscope operating at 120 kV, at a magnification of 52,000× that resulted in a pixel size of 2.13 Å at the specimen plane. Images were acquired with a Gatan 2 K CCD camera using a nominal defocus of 1500 nm at 10° tilt increments, up to 50°. The tilts provided additional particle orientations to improve the image reconstructions. Particles were picked semi-automatically using EMAN2[77] and put into a particle stack. Initial, reference-free, two-dimensional class averages were calculated and particles corresponding to complexes (with one, two, or three Fabs bound) were selected into a substack for determination of an initial model. The initial model was calculated in EMAN2 using threefold symmetry and EMAN2 was used for subsequent refinement using threefold symmetry. In total, 15,269 particles were included in the final reconstruction for the three-dimensional average of CH505 SOSIP.664 trimer complex with DH522.2. The resolution of the final model was determined using a Fourier shell correlation (FSC) cutoff of 0.5. The cryo-electron tomography structure of b12-bound gp120 trimer (PDB ID: 3DNL)[78] and crystal structure of DH564 were manually fitted into the EM density and refined by using the UCSF Chimera "Fit in map" function[79]. The FSC curve validates the resolution for the EM structure of the DH522.2-SOSIP.664 complex.

**Statistical analysis.** For macaques, we calculated the mean IGHV mutation frequency and HCDR3 length of CH505 differential and non-differential-binding antibodies, and non HIV-1-reactive antibodies, for each animal and then collectively compared the means across animals for different antibody groups using exact Wilcoxon test with FDR correction (SAS9.4). Exact Wilcoxon test was limited to comparisons with $N > 1$ for animals studied across groups, and for macaques with <2 Abs(5356, 5362), the isolated Abs were excluded from statistical analyses. For comparing properties of antibodies from ≥3 vaccine groups, Kruskal–Wallis test was applied (GraphPad Prism 7.0). Binding or neutralizing titers of mAbs induced by TF and sequential CH505 Envs were compared using exact Wilcoxon test (GraphPad Prism 4.0). A $P$ value <0.05 determined statistical significance.

For all mouse strains, immunization groups, and naive- or ex vivo-stimulated B cell subsets, means were calculated from using a minimum of four animals. Pairwise comparisons between WT B6 and various CH103 KI strains, between CH103 het UCA double KI vaccination groups, and between CH103 UCA double KI B cell subsets were all performed using a two-tailed Student's $t$-test (Microsoft Excel for Mac 2011, version 14.1.0). $P$ values of <0.05, <0.005, and <0.0001 determined increased significance levels. Regression measurement for strength of the inverse correlation between B cell developmental blockade and retention of Env specificity/receptor editing in CH103 UCA double KI mice was evaluated using Pearson correlation coefficient analysis (also in Excel v14.1.0).

**Data availability.** The data that support the findings of this study are available from the corresponding authors upon reasonable request. Coordinates and structure factors have been deposited in the Protein Data Bank under accession codes 5UKN (DH522UCA unliganded Fab), 5UKO (DH522IA unliganded Fab), 5UKP (DH522.1 unliganded Fab), 5UKQ (DH522.2 unliganded Fab), and 5UKR (DH522.2-gp120 complex). DH522 lineage Ab sequences have been deposited in Genbank under accession numbers MF848968-MF848975. The Electron Microscopy map of CH505 SOSIP in complex with DH522.2 has been deposited in the Electron Microscopy Data Bank with accession code EMD-9401.

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

## Acknowledgements

We thank Lawrence Armand for generating fluorophore-labeled gp120 Envs for flow cytometry, Ashley Trama for recombinant antibody and gp120 Env production, Duke Human Vaccine Institute (DHVI) programs and finance staff for project oversight, and Bob Bailer and the VRC Immunology core laboratory for performing the large panel assays. We also acknowledge the contributions of technical staff at the DHVI, including Melissa Cooper, Krissey E. Lloyd, Amy Wang, Tarra Von Holle, Alexis Sponaugle, Yousef Abuahmad, Maggie Barr, Callie Vivian, Stormi Chadwick, Giovanna Hernandez, James Pritchett, Erika Dunford, Meng Chen, Nelson Wu, and Andy (Shi) Wang. Use of the Advanced Photon Source was supported by the US Department of Energy, Office of Science, Office of Basic Energy Sciences, under contract no. W-31-109-Eng-38. This work was supported by grants from the NIH, National Institute of Allergy and Infectious Diseases, Division of AIDS; UM-1 grant for the Duke Center for HIV/AIDS Vaccine Immunology-Immunogen Discovery (CHAVI ID; UM1 AI100645), R01 grants AI087202, AI118571, and AI120801, and Duke University Center for AIDS Research (CFAR; P30-Al-64518).

## Author contributions

B.F.H. conceived and designed the study, evaluated all data, and wrote the paper; L.V. conceived and designed the mouse studies, analyzed the data, co-wrote, and edited the paper; W.B.W. designed and performed experiments for NHP studies, analyzed data, co-wrote, and edited the paper; F.G., J.Z. and C.J. designed and/or performed experiments for mice studies and analyzed data; N.I.N. performed X-ray crystallography studies; D.F. and S.C.H. performed negative stain electron microscopy studies; M.A.M., K.L., D.J.M. and J.F.W. performed flow cytometry sorts and/or isolated antibodies; T.B.K., A.R., K.W. and J.A.H. analyzed antibody gene sequences; T.B. performed NGS; N.V. performed statistical analyses; S.M.A., A.F. and R.P. characterized antibody-binding specificities; H.-X.L. and K.O.S. provided recombinant Envs and mAbs; D.C.M., M.B., M.L., J.R.M., S.-M.X. and A.E. performed neutralization assays and data analysis; S.S., R.S., L.S. and C.B. performed NHP immunizations and care; A.N. and H.B.-V. performed mice immunizations and care; H.B., X.N. and G.K. characterized antibody polyreactivity; S.G.R., C.B.F., and K.C. provided adjuvants; and M.K. and D.F. provided adjuvants and contributed to the development of the NHP study protocol.

## Additional information

**Competing interests:** S.G.R., C.B.F. and K.C. are employees of Infectious Disease Research Institute (Seattle, WA). M.K. and D.F. are employees of the GSK group of companies. The remaining authors declare no competing financial interests.

