## [Peer Review File · Nature Communications]

Reviewers' comments:

Reviewer #1 (Remarks to the Author):

The study discusses efforts to elicit CH103-like broadly neutralizing antibodies through a sequential immunization scheme with recombinant CH505-derived envelope immunogens. Immunizations were performed in non-human primates (NHP). The bottom line is that CH103-like antibodies were not elicited. However, the authors discuss in a very thoughtful manner, potential reasons for this and executed appropriate experiments to address this issue. This is a major strength of this study. Overall the study is outstanding and I do not have major concerns regarding the experiments or with the interpretation of the results. The underlying mechanisms that prevent the rapid elicitation of CH103-like neutralizing antibodies in the KI mice are well discussed. However, the study does not provide direct evidence that similar tolerance-related blockages are operational in the NHP case. The only direct evidence provided for the lack of elicitation of CH103 antibodies is the lack (infrequent?) of expression of appropriate VL genes in NHP.

Fig 3. The NHP antibodies (DH522.1 and DH522.2) neutralize heterologous viruses, while CH103UCA does not. HC/LC chimeric antibodies between the NHP antibodies and CH103 UCA could be generated and tested for neutralization, to better investigate the contribution of the HC and LCs in neutralization.

Fig 7a. TF+w53 immunization results in approximately a log higher TF Env-specific B cells (and in neutralizing antibody responses) than immunization with TF. Why is that? Does the w53 envelope 'rescue' somehow envelope+ B cells? What happens if only wk53 envelope is used alone as an immunogen in these heterozygous mice?

Reviewer #2 (Remarks to the Author):

The manuscript by Williams and colleagues addresses the question of why it is so difficult to elicit broadly neutralizing antibodies against HIV ENV, focussing on the conserved CD3 binding site.

The study focuses on antibodies related to CH103, a broadly neutralizing antibody lineage that arose during chronic HIV infection of an African individual, where the evolution of ENV variants and antibody variants has previously been longitudinally tracked and analysed in remarkable detail.

One line of experimentation attempts to reproduce the elicitation of CH103-like antibodies by sequentially immunizing rhesus macaques with ENV proteins corresponding to different timepoints in the natural history of the CH103 lineage. This proved not to elicit broadly neutralizing antibodies, but did generate antibodies related to CH103 but lacking the corresponding lambda light chain – possibly because the macaques lacked the necessary VLambda element present in humans.

The clearest conclusions come from the second line of experiments, where mice are engineered by gene targeting in their germline H and L chain genes with rearranged VDJ and VJ elements encoding the inferred unmutated common ancestor of the CH103 antibody. These experiments are elegant and conclusively demonstrate that the unmutated CH103 antibody undergoes strong negative selection.

There is lower surface IgM on nascent immature B cells in the bone marrow, editing of light chains, and further diminished surface IgM, calcium signalling and poor accumulation as transitional and mature B cells. These are all well established responses of developing B cells when their surface Ig binds too strongly to systemic self-antigens. The nature of the self-antigens responsible for negative selection of CH103 is not determined, but these findings alone provide the clearest evidence yet that CD4-binding site antibodies are subject to multiple mechanisms of self-tolerance and negative selection.

Env immunization experiments show that the CH103-expressing B cells can be induced to hypermutate and switch to IgG, albeit very inefficiently.

The results are for the most part compelling and will be of wide interest to readers interested in HIV, antibodies, vaccines and self-nonsel discrimination.

As detailed below, there are some technical and presentation issues that should be addressed to further improve an already very interesting manuscript.

1. Excessive use of sub-field specific acronyms makes the study very hard to read. Avoid abbreviating rhesus macaque to RM, or NHP (as done in the figures); avoid shortening knock-in to KI; bnAbs is ok but nAbs better to spell out neutralizing antibodies; etc. Also, figure legends need to explain better how the results were obtained, and avoid interpretation of the results. For example legend to Figure 5D does not explain that the results are from sorting single B cells regardless of Env binding and sequencing H and L chain mRNA. Legend to Figure 2A needs to clarify % of what: all leukocytes, B cells, or memory B cells?
2. Nature of immunogen. In the macaque and mouse studies, it is not explained whether the immunization used ENV trimers, monomers, SOSIPs, were the glycans native complex carbohydrates or high mannose or EndoH treated? The form of antigen given should be explained in the results, figure legend and in more detail in the methods.
3. In Figure 2, need to provide representative flow cytometric plots showing the gating strategy to enumerate ENV-binding memory B cells. Full gating strategy should be provided in supplementary figure. The conclusion at line 156 of the results "sequentially-administered Envs also induced higher frequencies of differential-binding memory B-cells" is not supported by statistical evidence, and there appears little difference in Figure 2b. In Figure 2c, please use same Y axis scale for all three panels to improve comparison of the regimes.
4. Affinity for Env. In the results text and legend for Figure 3, it is important to clarify that

deglycosylated Env was used for crystallography. When CD4bs antibodies have been crystallized with natively glycosylated Env, half the contact surface is between antibody and N-glycan's shrouding the binding site. What is known about the affinity of CH103 UCA for deglycosylated versus natively glycosylated Env?

Line 303 cites ref 6 where the K_d for (high mannose?) glycosylated Env was measured to be 36 nM. This was measured by SPR with divalent IgG, and may not reflect the monovalent affinity, which could be lower. Important to state the K_d , how it was measured, and whether it is the single-site affinity or bivalent avidity, in order to compare like with like vis other antibodies referred to here.

The Abstract states that the "non-edited precursors underwent limited affinity maturation" in the immunised mice. Results only show they underwent hypermutation, and the fraction of non-edited precursors among IgG+ cells appears comparable with repeated TF or sequential immunization. There is no convincing evidence that sequential immunization was better at recruiting unedited cells to hypermutate and switch.

Figure 5g. Given that CH103 likely makes extensive contacts with N-glycans around the CD4binding site, it is likely that it or the unmutated common ancestor has substantial affinity for N-glycans on a variety of self proteins. It would be valuable to test this, since low affinity binding to cell surface N-glycans would explain the downregulation of surface IgM and negative selection observed.

5. Figure 7 is the weakest part of the study. Need to show representative plots of IgG+ cells and gates used. Ideally would be good to show as IgG versus ENV TF binding. The text on line 339 states that non-differential Env binders "represent ~35% of the total IgG+ memory pool" but the plots gated on this pool show less than 2% ENV binders, most of which appear to be differential binders to native and not CD4-binding site mutant ENV.

The corresponding analysis of all B cells in Supp Fig 5 appears to have an error, because the saline injected mice should have ~8% ENV binders among all B cells (as shown in Figure 5A) yet there are none and the plots appear to have fewer events overall and be processed differently.

Figure 7D should be omitted. It "overcooks" the data in 7D, and one can't do statistics on ratio since not normally distributed because numerator and denominator are reciprocals. Just leave at C, which is a more accurate yet convincing portrayal.

In Figure 7E and F it is surprising that only 9 of 81 cells sorted based on binding Env but not CD4bs-mutant Env actually express the targetted H and L chain, and indeed 70% of the presumed CH103-expressing memory B cells lack the CH103 H chain. One would expect nearly 100% to be expressing CH103 H and L chain. Is this a technical problem with ENV staining and sorting in the immunized mice? It casts doubt on the conclusion that the increased frequency of differential Env binding IgG+ cells upon sequential immunization in Figure 7A, and the increased serum IgG binding to wildtype relative to mutant Env, represents increased activation of unedited CH103 cells. Most of the cells appear not to

employ the knocked in CH103 H chain, and only a minority of those that do employ the CH103 L chain. Does this mean the endogenous mouse repertoire is better equipped to mount IgG antibodies against the CD4 binding site? Needs discussion and clarification.

Reviewer #3 (Remarks to the Author):

The article by Williams et al. entitled 'Initiation of HIV Neutralizing B Cell Lineages with Sequential Envelope Immunizations' is a very thorough multi-center study describing the shaping of the immune response to HIV trimers in model systems with engineered B-cell lineages. The study represents a thorough exploration of a concept that many investigators in the field are interested and will therefore be of wide interest and importance.

Focusing on the crystallographic datasets corresponding to the reported structures of DH522.1, DH522UCA, DH522IA, DH522.2, and DH522.2-gp120 core. The data processing and refinement statistics are reported appropriately and all the datasets exceed expected minimum standards for the reported resolutions. The datasets are of suitable quality to reliably support the presented crystal structure models in Figure 3.

The cryo-EM data is of low resolution but the presented class averages of the DH522-CH505 SOSIP complex provide good supporting evidence of the inferred protomer neighbor clash of DH522 from the crystallographic studies. Given the low resolution and on-going variation in data processing methods adopted across the EM community, I would recommend including corresponding EM for all the antibodies studied crystallographically. This is particularly important given the low resolution and unusual conformation of the presented 'open' trimer.

We appreciate the reviewers' thorough and thoughtful comments and are glad they were so positive in their overall assessment of our study. We have answered all the questions from the reviewers, and where applicable, performed additional experiments requested. Our manuscript is now considerably stronger, and we now hope it is acceptable for publication in *Nature Communications*. For ease in review, the comments of the reviewers are numbered and listed below followed by our answers in blue font. The changes we have made are indicated in yellow in the manuscript.

Reviewer #1

The study discusses efforts to elicit CH103-like broadly neutralizing antibodies through a sequential immunization scheme with recombinant CH505-derived envelope immunogens. Immunizations were performed in non-human primates (NHP). The bottom line is that CH103-like antibodies were not elicited. However, the authors discuss in a very thoughtful manner, potential reasons for this and executed appropriate experiments to address this issue. This is a major strength of this study. Overall the study is outstanding and I do not have major concerns regarding the experiments or with the interpretation of the results.

1. The underlying mechanisms that prevent the rapid elicitation of CH103-like neutralizing antibodies in the KI mice are well discussed. However, the study does not provide direct evidence that similar tolerance-related blockages are operational in the NHP case. The only direct evidence provided for the lack of elicitation of CH103 antibodies is the lack (infrequent?) of expression of appropriate VL genes in NHP.

Answer: We agree with the reviewer that we did not show direct evidence in the macaque model for similar tolerance mechanisms observed in the elicitation of CH103-bnAb in the Knock-in mouse model. Studying these mechanisms in outbred macaques is difficult and current studies to date test these concepts only in knock-in mouse models. Our collaborator and co-author of this manuscript, Thomas Kepler, found that the closest gene segment found in the analysis of ten macaque genomes is 85% identical to IGLV3-1 (A. Ramesh et al., submitted) used by CH103. Because the

macaque Ig loci show high inter-individual diversity, we used next-generation sequencing (NGS) using IGLV3-specific primers to further probe macaque 5556 for germline gene segments closer to IGLV3-1. We reanalyzed our Illumina Next Generation Sequencing (NGS) dataset of IGLV3 genes to determine the sequence similarity of candidate germline IGLV genes from animal 5556, which generated CD4 binding-site neutralizing antibody with limited neutralization breadth to human IGLV3-1. We used statistical phylogenetics to infer germline gene segments directly from recombined mature variable region genes in NGS data. Using this new approach we found that the nearest likely germline sequence is about 85% identical to IGLV3-1. We have updated the manuscript to reflect these changes on page 10 (results) and the methods on page 8 of the supplemental online materials. Thus, our data suggested that insufficient macaque and human gene identity was a plausible reason for inability of macaques to make CH103-like bnAbs.

- 2.** Fig 3. The NHP antibodies (DH522.1 and DH522.2) neutralize heterologous viruses, while CH103UCA does not. HC/LC chimeric antibodies between the NHP antibodies and CH103 UCA could be generated and tested for neutralization, to better investigate the contribution of the HC and LCs in neutralization.

Answer: *As suggested by the reviewer, to test the contribution of the heavy (VH) and light (VL) chains in neutralization by DH522 and CH103 lineage antibodies, we generated VH/VL chimeric antibodies between DH522, bnAb CH103 and the lesser neutralizing CD4 binding-site antibody F105. These chimeric antibodies were tested for binding in ELISA and neutralization in TZM bl assay; the results of these assays are now reported in revised Figure 1f of the supplemental materials.*

We found that the chimeric antibodies with DH522VH and CH103VL did not bind CH505 TF gp120 Env or neutralize CH505.w4.3 virus. Only the chimeric antibodies bearing DH522VL demonstrated binding to CH505 gp120 Envs albeit weaker than CH103, DH522 and F105 mAbs. Additionally, the chimeric DH522VL antibodies did not neutralize autologous tier 2 CH505 or tier 1 CH505.w4.3 viruses nor demonstrated neutralization breadth of 6 heterologous viruses tested that were neutralized by CH103, DH522 or F105.

Inserted in the main text on page 8 is the following section describing these results: "Chimeric antibodies of heavy and light chain genes of DH522, CH103 and F105 improved neither binding nor neutralization breadth of DH522 and F105 (see Supplementary Fig. 1f), suggesting that compensatory mutations in CH103 heavy and light chain genes are necessary for mAb binding and neutralization."

3. Fig 7a. TF+w53 immunization results in approximately a log higher TF Env-specific B cells (and in neutralizing antibody responses) than immunization with TF. Why is that? Does the w53 envelope 'rescue' somehow envelope+ B cells? What happens if only wk53 envelope is used alone as an immunogen in these heterozygous mice?

Answer: Regarding why “TF+w53 immunization results in approximately a log higher TF Env-specific B cells (and in neutralizing antibody responses) than immunization with TF”, this is an important question. Although we do not have direct evidence for the mechanism behind this, we do not think week 53 Env directly rescues differential Env binding (lineage positive/unedited B-cells), since we do not think week 53 Env has high enough affinity for UCA B cells to activate anergic B-cells. The two possibilities we do favor, however, are that week 53 Env indirectly helps “break” anergy in double knock-in clones, either by:

a) diverting the non-lineage positive B-cell response, thus reducing competition with non-anergic clones. This would specifically occur via week 53 Env potentially cross-reacting more strongly than TF with either of two subsets: i) pre-immune knock-in light chain-edited (TF Env-) clones and/or ii) vaccine-induced knock-in heavy chain and light chain-edited (TF Env+ but non-differential) clones that start appearing after TF priming.

*b) preferential activation of the rare double knock-in positive clones initially activated by TF to switch and expand, because of their preferential (albeit limited) SHM induced (relative to heavy or light chain only knock-in clones; **Fig 7G**), thus resulting in a higher fraction of clones with affinity for, and thus further expansion by week 53 Env.*

These possibilities are now stated in the discussion on page 21 of the manuscript.

Regarding the related query: “What happens if only wk53 envelope is used alone as an immunogen in these heterozygous mice?”, this implies week 53 Env should directly activate the UCA in vivo. While favoring the indirect explanations we provide above for the observed increases in lineage-specific+ B-cells induced by the sequential addition of week 53 and TF Envs (relative to repeated TF immunization), this is certainly a control we plan on doing in our next vaccine iterations using this model, to formally rule out the direct week 53 Env-mediated activation of UCA⁺ B-cells. Although the logistics of doing such an experiment falls outside the scope of this manuscript (due to the amount of time required to get results from immunizing a new cohort), we do have two lines of evidence, that although not definitive in ruling this possibility out, are suggestive in arguing against it.

Reviewer #2

The manuscript by Williams and colleagues addresses the question of why it is so difficult to elicit broadly neutralizing antibodies against HIV ENV, focussing on the conserved CD3 binding site.

The study focuses on antibodies related to CH103, a broadly neutralizing antibody lineage that arose during chronic HIV infection of an African individual, where the

evolution of ENV variants and antibody variants has previously been longitudinally tracked and analysed in remarkable detail.

One line of experimentation attempts to reproduce the elicitation of CH103-like antibodies by sequentially immunizing rhesus macaques with ENV proteins corresponding to different timepoints in the natural history of the CH103 lineage. This proved not to elicit broadly neutralizing antibodies, but did generate antibodies related to CH103 but lacking the corresponding lambda light chain – possibly because the macaques lacked the necessary Vlambda element present in humans.

The clearest conclusions come from the second line of experiments, where mice are engineered by gene targeting in their germline H and L chain genes with rearranged VDJ and VJ elements encoding the inferred unmutated common ancestor of the CH103 antibody. These experiments are elegant and conclusively demonstrate that the unmutated CH103 antibody undergoes strong negative selection.

There is lower surface IgM on nascent immature B cells in the bone marrow, editing of light chains, and further diminished surface IgM, calcium signalling and poor accumulation as transitional and mature B cells. These are all well established responses of developing B cells when their surface Ig binds too strongly to systemic self-antigens. The nature of the self-antigens responsible for negative selection of CH103 is not determined, but these findings alone provide the clearest evidence yet that CD4-binding site antibodies are subject to multiple mechanisms of self-tolerance and negative selection.

Env immunization experiments show that the CH103-expressing B cells can be induced to hypermutate and switch to IgG, albeit very inefficiently.

The results are for the most part compelling and will be of wide interest to readers interested in HIV, antibodies, vaccines and self-nonself discrimination.

As detailed below, there are some technical and presentation issues that should be addressed to further improve an already very interesting manuscript.

- 4.** Excessive use of sub-field specific acronyms makes the study very hard to read. Avoid abbreviating rhesus macaque to RM, or NHP (as done in the figures); avoid shortening knock-in to KI; bnAbs is ok but nAbs better to spell out neutralizing antibodies; etc. Also, figure legends need to explain better how the results were obtained, and avoid interpretation of the results. For example legend to Figure 5D does not explain that the results are from sorting single B cells regardless of Env binding and sequencing H and L chain mRNA. Legend to Figure 2A needs to clarify % of what: all leukocytes, B cells, or memory B cells?

Answer: We thank the reviewer for the suggestions. We have edited the manuscript text to refer to rhesus macaques as macaques or NHPs, and nAb abbreviation was replaced with neutralizing antibodies. In figures where these abbreviations are in the figure, we indicate the abbreviation key in the figure legend. We have also revised our figure legends for clarity in how data were generated.

With respect to Figure 5D in particular, we assume the reviewer means Figure 5F (since sorting/cloning was only performed for data related to that panel). As per the reviewer's suggestion, that figure legend has now been revised to state that single cell sorts of total (unselected for Env binding) mature IgG⁺ B-cells were performed and that the VDJ/VJ pairs from single cells were recovered by RT-PCR. Furthermore, the legend of Figure 2A is now also revised, to read as follows: "One million peripheral blood mononuclear cells (PBMCs) from immunized RMs were phenotyped by FACS analysis for memory B cells that had CH505 Env differential binding. Shown is frequency (%) of memory B cells in PBMC that demonstrated CH505 Env differential binding at timepoints throughout the immunization schedule per animal (each line on graph)"

5. Nature of immunogen. In the macaque and mouse studies, it is not explained whether the immunization used ENV trimers, monomers, SOSIPs, were the glycans native complex carbohydrates or high mannose or EndoH treated? The form of antigen given should be explained in the results, figure legend and in more detail in the methods.

Answer: We thank the reviewer for the suggestions. As recommended, we outlined in more details the form of CH505 Envs used as immunogens in the methods of the supplemental materials, and were also more descriptive of the Env forms used in the results and figure legends by using the term CH505 gp120 Env monomer instead of CH505 Env. We included a recombinant Env gp120 expression section on page 3 of the supplemental materials. Previous work has demonstrated that the glycans on recombinant gp120 Envs are more complex than glycans on virus associated gp160 Envs (Doores et al. PNAS 2010; 107(31):13800-5). We have studied CH505 Env gp120 proteins expressed in 293F cells by glycan site-specific mass spectrometry; the glycans were found to be a mixture of complex and high mannose residues (Desaire H, Alam SM, Haynes BF et al., unpublished). This is now stated on page 4 of the supplemental materials.

6. In Figure 2, need to provide representative flow cytometric plots showing the gating strategy to enumerate ENV-binding memory B cells. Full gating strategy should be provided in supplementary figure. The conclusion at line 156 of the results "sequentially-administered Envs also induced higher frequencies of differential-binding memory B-cells" is not supported by statistical evidence, and

there appears little difference in Figure 2b. In Figure 2c, please use same Y axis scale for all three panels to improve comparison of the regimes.

Answer: *We agree with the reviewer that the conclusion on line 156 is not statistically supported. We have rephrased the sentences on page 7 of the manuscript to say “Among macaques immunized with TF gp120 Env, 11% of the CH505 Env-reactive antibodies were CH505 differential binders; however, among those immunized with sequential combinations of CH505 gp120 Envs, 16% were CH505 differential binders (Figure 2b). Thus, we observed a trend for animals immunized with sequential CH505 gp120 Envs to make more CH505 differential-binding antibodies, albeit not statistically significant.”*

As recommended, Figure 2C has been edited to show the same Y axis scale for all three panels, and a new Supplementary Figure 2 shows representative flow cytometric plots with the gating strategy to enumerate Env-binding memory B cells. In the new Supplementary Figure 2, we showed the representative flow cytometry plots of CH505 differential binding memory B cells above background, at 2 weeks post 3rd immunization for animals immunized with CH505 TF gp120 Env monomer alone (N=4) and sequential combinations of CH505 gp120 Env monomers (N=8). Most immunized animals had CH505 differential binding memory B cells at 2 weeks post 3rd immunization.

7. Affinity for Env. In the results text and legend for Figure 3, it is important to clarify that deglycosylated Env was used for crystallography. When CD4bs antibodies have been crystallized with natively glycosylated Env, half the contact surface is between antibody and N-glycan's shrouding the binding site. What is known about the affinity of CH103 UCA for deglycosylated versus natively glycosylated Env?

Answer: *We agree with the reviewer that we should clarify that deglycosylated Envs were used for x-ray crystallographic studies. Negative stain electron microscopy studies were performed with fully glycosylated Envs. We have edited the legend of Figure 3E on page 31 to reflect these changes.*

It was previously reported that glycan-deleted trimers, including CH505 DS SOSIP.ΔGly4, showed increased binding to CH103 (Zhou et al. Cell Reports 19, 2017;719-732), consistent with previous reports that glycans around the CD4 binding site occlude antibody access to the CD4 binding-site epitopes (Stewart-Jones et al. Cell 165, 2016;813-826). Furthermore, macaque immunizations with CH505 DS SOSIP.ΔGly4 generally induced high titer autologous neutralizing antibodies to the deglycosylated CH505 trimer (Zhou et al. Cell Reports 19, 2017;719-732), but the induction of blood-derived CH103-like antibodies in those macaques were not studied.

In this study, we used natively glycosylated CH505 gp120 Envs as immunogens. To address the reviewer's question about the affinity of CH103 UCA for deglycosylated versus natively glycosylated Env, we first determined if we could detect differences in CH103 and CH103UCA binding in ELISA to CH505 TF gp120 and CH505 TF gp120 mutant (CH505 TF gp120 N276DN463D) with deleted glycans around the CD4 binding site as previously reported (Zhou et al. Cell Reports 19, 2017;719-732). We screened DH522 lineage Abs for comparison. We found that the CH103 and DH522 lineage Abs bound CH505 TF gp120 and CH505 TF gp120 N276DN463D Envs equally well, demonstrating that the CH103UCA and other Abs tested have a similar affinity for the glycosylated and deglycosylated gp120 Envs evaluated. Additionally, we tested these antibodies for neutralization of pseudoviruses bearing wildtype CH505 or 426c virus strain Envs and mutants with deletion of the glycans in the vicinity of the CD4 binding site. We found that CH103 demonstrated improved neutralization titers against HIV-1 strains bearing Envs with glycan deletions compared to the viruses bearing natively glycosylated Envs. These results are reported on pages 8-9 of the manuscript and shown in new Supplementary Figure 3 on page 29 of the supplemental materials. These data suggest that unlike gp120 monomers, glycosylation of Env trimers may impact the neutralization capacity of the UCA and mature CH103 lineage antibodies.

8. Line 303 cites ref 6 where the Kd for (high mannose?) glycosylated Env was measured to be 36 nM. This was measured by SPR with divalent IgG, and may not reflect the monovalent affinity, which could be lower. Important to state the Kd, how it was measured, and whether it is the single-site affinity or bivalent avidity, in order to compare like with like vis other antibodies referred to here.

Answer: *The sentence in question is on line 328-329 of the revised manuscript – “TF Env gp120 binds the CH103 UCA IgG Ab in vitro more tightly (KD in the high nM range (6)) than activation thresholds estimated for triggering naïve B-cells in vivo (38-40). We previously reported that CH103UCA bound to Env gp140 trimer with a bivalent Kd value of 36 nM (Liao et al. Nature 2013;496(7446):469-76). We have measured the monomeric gp120 binding affinity for the CH103UCA to be 521 nM (Alam SM, Haynes BF et al., unpublished), which is higher than the threshold for BCR signaling (Shih et al. Nature Immunology 2002;3(6):570-5). Thus, these findings are consistent with our statement on line 328-329 of the manuscript. The affinity measurements for monomeric CH505 gp120 Envs and CH103 lineage Abs were reported on page 5 of the supplemental materials.*

9. The Abstract states that the “non-edited precursors underwent limited affinity maturation” in the immunised mice. Results only show they underwent hypermutation, and the fraction of non-edited precursors among IgG+ cells appears comparable with repeated TF or sequential immunization. There is no

convincing evidence that sequential immunization was better at recruiting unedited cells to hypermutate and switch.

Answer: We agree with this comment by the reviewer. Thus, we have re-worded the abstract (see revised lines 33-34) in order to more clearly reflect this important distinction.

10. Figure 5g. Given that CH103 likely makes extensive contacts with N-glycans around the CD4 binding site, it is likely that it or the unmutated common ancestor has substantial affinity for N-glycans on a variety of self-proteins. It would be valuable to test this, since low affinity binding to cell surface N-glycans would explain the downregulation of surface IgM and negative selection observed.

Answer: We thank the reviewer for this comment. See answer to question #7. In addition, to determine if the CH103UCA binds to N-glycans, we performed a glycan binding array with CH103UCA and for comparison included CH103 and the DH522 lineage Abs. We found that neither the CH103UCA nor the other Abs tested unambiguously bound N-glycans. V3-glycan targeted Abs PGT128 and DH501 were used as positive controls. Thus, the glycans around the CD4 binding-site appear to block CH103 lineage Ab binding, while the Abs do not interact directly with glycans. These data are reported on page 9 of the manuscript and shown in a new Supplementary Figure 3c on page 29 of the supplemental materials.

11. Figure 7 is the weakest part of the study. Need to show representative plots of IgG+ cells and gates used. Ideally would be good to show as IgG versus ENV TF binding. The text on line 339 states that non-differential Env binders “represent ~35% of the total IgG+ memory pool” but the plots gated on this pool show less than 2% ENV binders, most of which appear to be differential binders to native and not CD4-binding site mutant ENV.

Answer: To address the reviewer’s point re: data presented in Figure 7A, we have now added representative plots to show the gating scheme in the supplemental material (Now Fig S8B). We have also shown IgG versus Env TF binding for the immunized groups, as an alternative way of representing the data (now revised Fig S8C). With respect to the latter however, we note that representing the data this way does not allow the differential binding IgG+ fraction (enriched for lineage-specific dKI+ clones) to be visualized, relative to the non-differential Env-binding IgG+ fraction (i.e. non-lineage specific, aka KI HC negative and LC negative cross-reactive clones, that are distinct from the Env- “LC only”-edited clones). Because we believe these binding schemes are important as comparison, specifically in order to reveal the third, distinct vaccine-specific population that gets expanded, we have respectfully elected to keep it this way in Figure 7A panel. Furthermore, showing IgG vs. tetramer gating (rather than first

gating on IgG), additionally obfuscates visualization of the extremely infrequent, yet more relevant Env+ IgG+ populations (i.e. this subset represents $\leq 2\%$ of the total IgG+ population) in relation to the much more predominant TF Env+ population seen in unswitched cells of immunized -see Supplementary Figs 8C and D in revised version); for this reason, we have also elected to keep the current figure (gated on IgG first) as part of the main Figure 7A panel.

Regarding the text on line 339 (now line 364-365 in the revised manuscript), we meant to indicate the percentage of non-differential binders that specifically fall within all TF Env-specific IgG+ B-cells (rather than the fraction of non-differential binders within the total IgG+ pool, which would be 35% of $2\%=0.7\%$). This 35% non-differential pool (within the TF Env-specific IgG+ subset) was calculated by subtracting the TF Env “differential-binding” IgG+ pool (bottom right panel) from the total TF Env-binding IgG+ pool (top right panel). To clarify this is what we intended to say in the results, we have re-worded the text accordingly in lines 364-369 of the revised manuscript text.

- 12.** The corresponding analysis of all B cells in Supp Fig 8 appears to have an error, because the saline injected mice should have $\sim 8\%$ ENV binders among all B cells (as shown in Figure 5A) yet there are none and the plots appear to have fewer events overall and be processed differently.

Answer: *“Supp Fig 8” referred to by the reviewer was Supplementary Fig 5. This is now Supplementary Fig. 8 on page 35 of the revised supplemental materials. The data shown in Supplementary Figure 8B is from heterozygous double knock-in mice (which to more closely mimic the physiological setting of clonal competition, was the version used in all immunization studies). In contrast, the data shown in Figure 5A is from homozygous double knock-in mice (which were used along with the other CH103UCA knock-in versions for initial characterizations pre-immunization, i.e. in naive strains). The considerable lowering of Env+ cells in the heterozygous versions of mice is most likely due to the overall increased permissiveness for receptor editing when the alternate unrearranged heavy and light chain alleles are available; as such, the heterozygous double knock-ins have nearly no Env+ mature B-cells as per the graph shown in Figure 5E). To emphasize this subtle yet important distinction, we now mention in the revised accompanying Supplementary Fig. 8 legend that: a) heterozygous mice are shown, and b) a back-reference to Fig 5E.*

However, we agree that while the frequency of TF Env+ B-cells in the IgM+ fractions from saline-administered mice is also lower than those in the other groups in het/het mice, the frequencies shown in the het/het mice (0.1% total cells) are considerably lower than some of the other het/het mice we have analyzed (the expansion of the predominant IgM+ TF-specific subset is not proportionally as large as the IgG+ subset in the immunized groups, relative to both GLA-SE or saline administration). For this reason, we have gone back and re-checked genotypes of all

het/het mice where we observe lower Env+ frequencies of total B-cells, and found that while the one shown in the original Supplementary Fig 5B is a bone fide “outlier” (variation possibly due to older ages and potential expansion by environmental antigens in some but not older mice, since saline administered are on the average older than naïve mice), two other mice in this category were indeed mis-genotyped as het/het (when in fact they were VDJ+/- knock-ins). To correct this error, we have i) included a representative animal of the mean found for total cells in the saline group, to replace the one shown in original Supplementary Fig. 5B (now supplementary Fig. 8C in the revisions), and ii) we have excluded the data points from the Fig 5E analysis, resulting now in higher overall frequencies (~15% and ~2%) of TF Env+ B-cells in the transitional and mature B-cell fractions of naïve het/het mice, respectively. The revised Fig 5E panel is now included, as well as edits to the results text to reflect this re-analysis. We regret the error in genotyping, but are grateful for the reviewer’s questions on this point.

- 13.** Figure 7D should be omitted. It “overcooks” the data in 7D, and one can't do statistics on ratio since not normally distributed because numerator and denominator are reciprocals. Just leave at C, which is a more accurate yet convincing portrayal.

Answer: *We agree, and have removed figure 7D in the revised version of our manuscript, in response to this specific suggestion.*

- 14.** In Figure 7E and F it is surprising that only 9 of 81 cells sorted based on binding Env but not CD4bs-mutant Env actually express the targetted H and L chain, and indeed 70% of the presumed CH103-expressing memory B cells lack the CH103 H chain. One would expect nearly 100% to be expressing CH103 H and L chain. Is this a technical problem with ENV staining and sorting in the immunized mice? It casts doubt on the conclusion that the increased frequency of differential Env binding IgG+ cells upon sequential immunization in Figure 7A, and the increased serum IgG binding to wildtype relative to mutant Env, represents increased activation of unedited CH103 cells. Most of the cells appear not to employ the knocked in CH103 H chain, and only a minority of those that do employ the CH103 L chain. Does this mean the endogenous mouse repertoire is better equipped to mount IgG antibodies against the CD4 binding site? Needs discussion and clarification.

Answer: *This is a good point that gets at the heart of the biology involved with HCDR3-binder CD4bs-specific bnAb lineage responses. We agree it is surprising that only 9/81 Env+ clones retain both their HC and LC KI alleles. We believe the answer for this at least in part reflects a key biological roadblock (extreme counter-selection for bona fide lineage-specific (KI HC+LC) clones, in favor of cross-reactive/off-target responses, even in the setting of KI model, despite having nowhere near the same degree of diversity,*

and thus repertoire competition as a fully polyclonal, non-KI system). Importantly however, we contend that despite the inefficiency with which differential binders are induced (and detected by the sorts), leads us to speculate that “sequential immunizations induces increased frequencies of lineage+ clones, found within the differential Env binding fraction of IgG⁺ cells (the major source of the reviewer’s concern). This is because despite a majority of clones being off-target, and isoleucine 371 likely being required, but not sufficient as a diagnostic for lineage+ clones, the important point is that significant enrichment for the bona fide lineage+ (knock-in heavy and light chain-expressing) clones in Env+ populations induced in immunized mice still remain, relative to those in the total IgG unimmunized pool. Thus, based on the specific data presented, there is relative >4 fold enrichment of bona fide knock-in heavy and light chain-expressing clones in differential-binding gates (9/81 = 11.1% in the four immunized mice, vs. 1/39 = 2.6% in the total IgG gate in the control-immunized mouse) demonstrates selection for the knock-in light chain by immunization. By extension, it then follows, given such differential-binders are induced at higher frequencies in sequential TF and week 53 Env immunized mice (Fig 7B), but enrichment for knock-in heavy and light chain positive clones is equivalent in both immunization regimens (~10% vs 12%; Fig 7E), that sequential immunization induces higher absolute numbers of such lineage-specific (knock-in heavy and light chain positive) clones per spleen (~3 per saline-immunized mouse, ~20 per TF (x2)-immunized mouse, and ~360/ TF and week 53 Env sequentially-immunized mouse).

With respect to the reviewers’ last query then, “Does this mean the endogenous mouse repertoire is better equipped to mount IgG antibodies against the CD4 binding site?”, the answer is definitely yes. To emphasize our explanation to this query, we have expanded on the model provided in Paragraph 6 of the discussion, based on the following reasoning: the Env immunogen is a highly complex antigen that likely has many immunodominant non bnAb epitopes (even closed trimers). Coupled with the fact the het dKI models can generate a diverse competing non-nergic (non-KI) B-cell repertoire (as is evident by the large # of endogenous HC and LCs utilized, comprising ~90% of the initial heterozygous repertoire; (Fig 7D), this predicts that there will be a high likelihood the larger pool of mostly non-nergic cells (which have an advantage for selection into GC and long-lived memory compartments), will predominate. This prediction is borne out by the kinetics of the response: there initially is a minor on track (differential-binding, i.e. lineage-enriched) “blip” seen at the plasma IgG and memory B-cell levels, but the plasma IgG response then goes off-track upon subsequent immunizations (Fig 7C). Thus, these findings suggest that the more responsive non-nergic repertoire initially wins out, and eventually, completely takes over peripheral immune constraints. As we state in our concluding paragraph, decreasing clonal competition (by iteratively focusing responses with an improved sequence of immunogens and or providing the anergic lineage-specific clones a more level playing

field, by immunomodulation of pro-survival / anti-apoptotic signals) would be predicted to enhance their development.

Reviewer #3:

The article by Williams et al. entitled 'Initiation of HIV Neutralizing B Cell Lineages with Sequential Envelope Immunizations' is a very thorough multi-center study describing the shaping of the immune response to HIV trimers in model systems with engineered B-cell lineages. The study represents a thorough exploration of a concept that many investigators in the field are interested and will therefore be of wide interest and importance.

Focusing on the crystallographic datasets corresponding to the reported structures of DH522.1, DH522UCA, DH522IA, DH522.2, and DH522.2-gp120 core. The data processing and refinement statistics are reported appropriately and all the datasets exceed expected minimum standards for the reported resolutions. The datasets are of suitable quality to reliably support the presented crystal structure models in Figure 3.

15. The cryo-EM data is of low resolution but the presented class averages of the DH522-CH505 SOSIP complex provide good supporting evidence of the inferred protomer neighbor clash of DH522 from the crystallographic studies. Given the low resolution and on-going variation in data processing methods adopted across the EM community, I would recommend including corresponding EM for all the antibodies studied crystallographically. This is particularly important given the low resolution and unusual conformation of the presented 'open' trimer.

Answer: We appreciate the reviewers' comments on the structural data. The recommendation about additional EM data was considered; however, the primary finding of the set of crystal structures for the unliganded Fabs in the DH522 lineage is that they are virtually identical in their folds. The antibodies in this lineage exhibited low levels of mutation from germline, none of which affected gross fold or CDR conformations. Thus, additional EM micrographs would not provide new information. These data can be found in a new figure (Supplementary Fig. 4) on page 30 of the supplemental online materials. We have a detailed explanation of these data on pages 10 of the supplemental materials.

REVIEWERS' COMMENTS:

Reviewer #1 (Remarks to the Author):

The authors appropriately addressed my queries in their response and updated their manuscript. I have no additional comments.

Reviewer #2 (Remarks to the Author):

The authors have carefully addressed each of the points raised at last review. It is an excellent and important study.

Reviewer #3 (Remarks to the Author):

Williams and colleagues should be commended on their timely and thorough study. The responses to all the reviewers comments seem thoughtful and appropriate. I have no hesitation in supporting this study for publication.